# WDR31 displays functional redundancy with GTPase-activating proteins (GAPs) ELMOD and RP2 in regulating IFT complex and recruiting the BBSome to cilium

Sebiha Cevik[1], Xiaoyu Peng[2], Tina Beyer[3], Mustafa S Pir[1], Ferhan Yenisert[1], Franziska Woerz[3], Felix Hoffmann[3], Betul Altunkaynak[1], Betul Pir[1], Karsten Boldt[3], Asli Karaman[5], Miray Cakiroglu[5], S Sadik Oner[4,5], Ying Cao[2], Marius Ueffing[3], Oktay I Kaplan[1]

The correct intraflagellar transport (IFT) assembly at the ciliary base and the IFT turnaround at the ciliary tip are key for the IFT to perform its function, but we still have poor understanding about how these processes are regulated. Here, we identify WDR31 as a new ciliary protein, and analysis from zebrafish and *Caenorhabditis elegans* reveals the role of *WDR31* in regulating the cilia morphology. We find that loss of WDR-31 together with RP-2 and ELMD-1 (the sole ortholog ELMOD1-3) results in ciliary accumulations of IFT Complex B components and KIF17 kinesin, with fewer IFT/BBSome particles traveling along cilia in both anterograde and retrograde directions, suggesting that the IFT/BBSome entry into the cilia and exit from the cilia are impacted. Furthermore, anterograde IFT in the middle segment travels at increased speed in *wdr-31;rpi-2;elmd-1*. Remarkably, a non-ciliary protein leaks into the cilia of *wdr-31;rpi-2;elmd-1*, possibly because of IFT defects. This work reveals WDR31–RP-2–ELMD-1 as IFT and BBSome trafficking regulators.

## Introduction

Cilia are structurally and functionally distinct cellular projections, consisting of a microtubule-based axoneme extending from a centriole-derived basal body anchored at the plasma membrane. Cilia consist of multiple sub-compartments (basal body, transition zone, and ciliary tip) that display different protein compositions and structures (Rosenbaum & Witman, 2002; Satir & Christensen, 2007; Blacque & Sanders, 2014). Motile cilia mediate the movement of unicellular organisms such as *Chlamydomonas reinhardtii* or are involved in fluid movement across a tissue surface (Sleigh, 1989; Silflow & Lefebvre, 2001). Non-motile cilia, also known as primary

cilia, possess mechanosensory, chemosensory, and osmosensory functions and coordinate a range of extrinsic signaling pathways involved in cellular behavior, tissue development, and homeostasis such as those mediated by Hedgehog (Hh), Wnt, and receptor tyrosine kinase ligands (e.g., PDGFα) (Scholey, 2007; Bloodgood, 2009; Nachury, 2014; Anvarian et al, 2019).

The relation between cilia and human disorders has led to a greater understanding of their importance for human health. Both motile and primary cilia have been linked to the heterogeneous class of diseases known as ciliopathies, including Joubert Syndrome, Meckel Syndrome (MKS), and Nephronophthisis (NPHP). Owing to the presence of cilia on most cell types, ciliary defects result in varying multi-organ phenotypes such as kidney defects, retinitis pigmentosa, pancreatic cysts, hearing loss, congenital heart disease, and polydactyly (Reiter & Leroux, 2017; Genomics England Research Consortium et al, 2019).

Over the last 20 yr, there has been a large effort to reveal the molecular composition of cilia and its sub-compartments using several independent approaches, including clinical genomics, proteomics, functional genomics, and bioinformatics (Avidor-Reiss et al, 2004; Li et al, 2004; Blacque et al, 2005; Arnaiz et al, 2009; Piasecki et al, 2010; Choksi et al, 2014; Mick et al, 2015; Jensen et al, 2016; Lambacher et al, 2016; Shaheen et al, 2016; UK10K Rare Diseases Group et al, 2016; Sigg et al, 2017; Breslow et al, 2018; Ruiz García et al, 2019; van Dam et al, 2019; Shamseldin et al, 2020). Many proteins that make up the cilium, and many proteins that regulate cilia biology, have been identified. These collective efforts have resulted in the identification of 302 genes that are certain to be involved in cilia biogenesis, and over 180 ciliopathy genes. CiliaCarta estimates the total number of the ciliary genes to be about 1,200 genes, implying that many more ciliary proteins and ciliopathy genes are yet to be discovered. Indeed, the genetic diagnosis of many ciliopathy disorders is still unknown (SYSCILIA Study Group et al, 2013; van Dam et al, 2019; Genomics England Research Consortium et al, 2019; Shamseldin et al, 2020, 2020).

[1]Rare Disease Laboratory, School of Life and Natural Sciences, Abdullah Gul University, Kayseri, Turkey   [2]School of Life Sciences and Technology, Tongji University, Shanghai, China   [3]Institute for Ophthalmic Research, Centre for Ophthalmology, University of Tuebingen, Tuebingen, Germany   [4]Goztepe Prof. Dr. Suleyman Yalcin City Hospital, Istanbul, Turkey   [5]Science and Advanced Technology Application and Research Center, Istanbul Medeniyet University, Istanbul, Turkey

Correspondence: oktay.kaplan@agu.edu.tr

Structural components of cilia and cilia cargos must be transported to the cilia to construct and sustain cilia, and mutations in genes involved in ciliary trafficking are commonly seen in ciliopathies. Cilia have a one-of-a-kind protein delivery system called intraflagellar transport (IFT). IFT is made up of multi-subunit protein complexes that travel bidirectionally along the cilia. The IFT complex contains two sub-complexes, IFT-A and IFT-B, consisting of 6 and 16 protein subunits, respectively (Prevo et al, 2017). The IFT-B sub-complex and Kinesin-2 motors mediate the motility of IFT and IFT cargos from the base of the cilia to the tip of the cilia (anterograde IFT), whereas the IFT-A sub-complex and the cytoplasmic dynein-2 motor facilitate the retrograde IFT transport (from the cilia tip to the base of the cilia) (Rosenbaum & Witman, 2002; Blacque, 2008). The IFT-A is involved in the transport of certain membrane proteins into the cilia (Lee et al, 2008; Mukhopadhyay et al, 2010; Liem et al, 2012). Mutations in genes encoding IFT components lead to defects in cilia formation in all examined organisms, indicating the importance of IFT for cilia assembly (Pazour et al, 2000; Prevo et al, 2017).

Bardet–Biedl syndrome (BBS) was classified as a ciliopathy in 2003, and eight of the highly conserved BBS proteins (BBS1, BBS2, BBS4, BBS5, BBS7, BBS8, BBS9, and BBIP10) establish a stable protein complex called the BBSome that undergoes IFT. Work from a range of organisms implicates the BBSome in a variety of cilia-related processes including acting as a cargo adaptor for removing proteins from cilia and as a regulator of the assembly and stability of IFT trains (Ansley et al, 2003; Ou et al, 2005, 2007; Nachury et al, 2007; Loktev et al, 2008; Lechtreck et al, 2009; Wei et al, 2012; Williams et al, 2014; Xu et al, 2015; Ye et al, 2018; Nozaki et al, 2019). For example, in the nematode *Caenorhabditis elegans* mutants lacking *bbs-7* or *bbs-8*, detachment of IFT-A and IFT-B in the amphid cilia was reported in the anterograde direction (Ou et al, 2005). Analysis with a hypomorphic mutant (*bbs-1*) revealed that BBSome is involved in attaching IFT-B components to the retrograde IFT machinery at the ciliary tips in the channel cilia of *C. elegans* (Wei et al, 2012). Though it is known that the BBSome governs the IFT assembly and returns of IFT from the cilia, we do not yet know which additional regulators control IFT assembly at the ciliary base and IFT turnaround at the ciliary tip.

To identify new IFT regulators, we focused on our single-cell RNA-seq data that compared expression profiles of ciliated cells with those of non-ciliated cells in the nematode *C. elegans*, and this work revealed novel cilia genes and potential IFT regulators, including *wdr-31* (Pir et al, 2023 Preprint). Our gene discovery approach in combination with the use of in vivo IFT microscopy analysis identified WDR31 (WD repeat domain 31) and two GAPs as IFT regulators that facilitate the ciliary entry of the IFT complex. Our work from zebrafish and *C. elegans* provides significant insight into the function of *WDR31* in cilia biogenesis. First, our work from *C. elegans* revealed that WDR-31 functions redundantly with two GAPs ELMD-1 (the sole ortholog of the human ELMOD proteins) and RPI-2 (human retinitis pigmentosa 2 orthologue) to control the cilia morphology. Second, knocking out *wdr-31* along with *elmd-1* or *elmd-1;rpi-2* causes IFT trafficking to be disrupted, resulting in ciliary tip accumulations of IFT-B components and the OSM-3/KIF17 motor, and significantly reduced BBSome recruitment to cilia. Third, fewer IFT/BBSome particles travel along the cilia in both anterograde and retrograde directions in *wdr-31;rpi-2;elmd-1* triple mutants, indicating that the IFT/BBSome entry into and exit from cilia is affected. Third, anterograde IFT in the middle segment moves faster in *wdr-31;rpi-2;elmd-1*. Finally, TRAM-1, a non-ciliary membrane protein, penetrates *wdr-31;rpi-2;elmd-1* cilia. Taken together, this work identifies WDR31 and two GAP proteins (ELMOD and RP2) as regulators for IFT and BBSome trafficking.

# Results

## WDR31 and ELMOD are evolutionarily conserved constituents of cilia

As part of our ongoing effort to reveal novel cilia genes, using single-cell RNA-seq data, we conducted comparative expression analysis in *C. elegans* and discovered that the WDR31 and ELMOD orthologues, *wdr-31* (T05A8.5) and *elmd-1* (C56G7.3), are likely expressed exclusively in ciliated sensory neurons (manuscript in preparation). The head (amphid) and tails (phasmid) of *C. elegans* contain a total of 60 ciliated sensory neurons, and the expression patterns of genes can be reliably analyzed using fluorescence-tagged promoter markers. Consistent with our predictions, promoter-based GFP reporters revealed that both *wdr-31* and *elmd-1* are expressed in most of the ciliated sensory neurons (Fig S1A). To explore if both proteins are concentrated within cilia, we investigated their subcellular localization in *C. elegans* and created CRISPR/Cas9-mediated knock-in of WDR-31::GFP and a transgenic animal expressing GFP-tagged ELMD-1. The endogenously expressed WDR-31::GFP is concentrated at the ciliary base in both head and tails, where it colocalizes with the basal body marker γ-tubulin (TBG-1, the ortholog of human TUBG1), and the IFT-140 (human IFT140) basal body signal (Fig 1A–D), so our co-localization data suggest that WDR-31 is a new cilia-associated protein. ELMD-1 is the sole orthologue of the three human ELMOD proteins (ELMOD1-3) that function as GAP. We found that GFP::ELMD-1 localizes to the periciliary membrane compartment (PCMC) and the basal body (BB) (marked with IFT-140), but is proximal to the MKS-6-labeled transition zone that is adjacent to the BB (Fig 1E–G) (Kaplan et al, 2012).

To determine if both human WDR31 and ELMOD3 (also known RBED1, the top blastp hit for *elmd-1*) are concentrated within cilia in mammalian cell lines, we generated hTERT-RPE1 cells co-expressing WDR31-RFP and ELMOD3-CFP and stained them with a ciliary marker ARL13B. Furthermore, WDR31-RFP or ELMOD3-CFP was transiently transfected into hTERT-RPE1 cells, and they were additionally stained for ARL13B. Our super-resolution microscopy analysis revealed that both WDR31 and ELMOD3 are enriched in the cilium (Figs 2A and B and S2B and C). Taken together, our complementary approach demonstrates that both WDR31 and ELMOD3 are evolutionarily conserved proteins associated with cilia.

## Wdr31 regulates ciliogenesis in the zebrafish ear

We turned our interest into zebrafish to develop an in vivo model for WDR31 to examine the role of WDR31. We first used the whole-

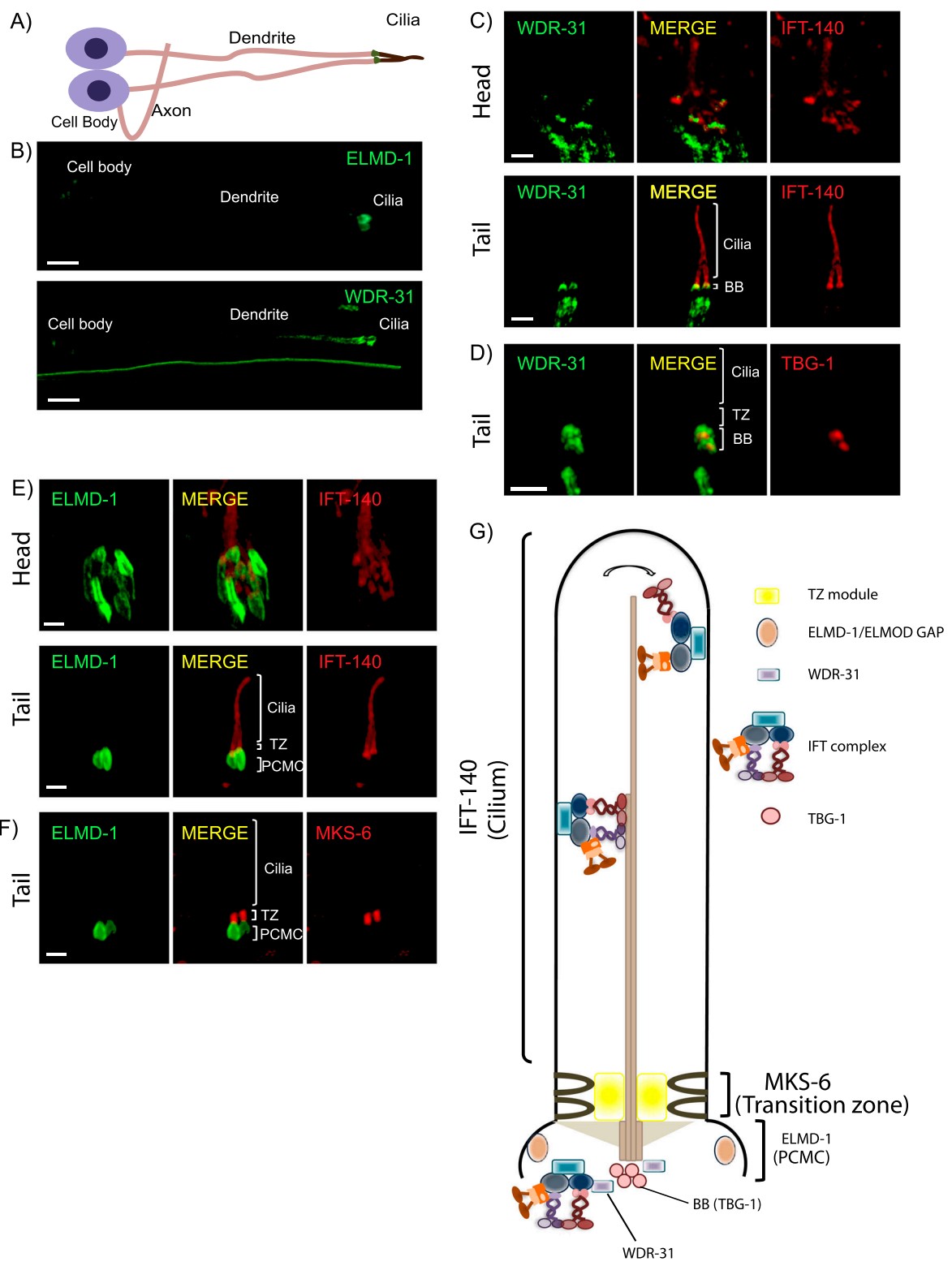

**Figure 1. WDR-31/WDR31 and ELMD-1/ELMOD proteins are evolutionary conserved ciliary proteins.**
**(A, B)** Shown are the representative drawing of the PHA/PHB sensory neuron (phasmid neurons located in the tail). Cilia, dendrites, axons, and cell soma (cell body) are depicted in the drawing. Fluorescence images from the transgenic strain carrying WDR-31::GFP or GFP::ELMD-1 were displayed in the PHA/PHB sensory neurons. Scale bars: 3 μm. **(C, D)** Co-localization of WDR-31::GFP (Green) with the IFT-140::mCherry (Red, an IFT-A component, a ciliary marker) or TBG-1:mKate (Red, γ-Tubulin, the basal body) in the tail (phasmids) and head (amphid) sensory neurons. **(E, F)** TZ and BB denote the transition zone and the basal body, respectively (E, F) GFP::ELMD-1 (Green) localizes

mount ISH within zebrafish to visualize the expression pattern of *wdr31* during zebrafish embryo development up to 1 day post-fertilization. The dynamic and wide expression of *wdr31* during zebrafish embryonic development (2-cell and 8-somite stages [SS]) has become restricted to the otic vesicle and brain region at 24 h post-fertilization (24 hpf) (Fig 3A). We next knocked out (KO) *wdr31* in zebrafish with the CRISPR/Cas9 system and the embryos showed heart edema and otolith malformation. As *wdr31* is expressed in the otic vesicle and the otolith development requires cilia, we checked the cilia in the otic vesicle. The cilia bundle in the lateral crista (LC) of the otic vesicle was stained with acetylated tubulin in WT and *wdr31* KO, and although we found that the length of LC cilia was comparable with WT, the width of the cilia bundle was reduced by 33% in LC of the otic vesicle compared with that in the control embryos, suggesting the cilia number is decreased (Fig 3B–D). Our zebrafish work indicates that Wdr31 plays a role in ciliogenesis in the zebrafish ear.

### Functional redundancy of WDR31, ELMOD, and RP2 for determining cilia morphology

We next wanted to gain mechanistic insight into the functions of *wdr-31* in cilia biogenesis. To this end, we employed the nematode *C. elegans* and obtained/generated three *wdr-31* alleles: *tm10423* (160-bp frameshift causing deletion), *syb1568* (1,888-bp deletion removing all exons except exon I), and *tur003* (1,276-bp deletion removing a large portion of exon II and exons III, IV, and exon V) (Fig S3B). In *C. elegans*, the lipophilic fluorescent dye-uptake assay is employed to indirectly evaluate the cilia structure, and our analysis revealed *wdr-31* mutants display WT level dye-uptake in both head (amphid) and tail (phasmid), suggesting cilia structures are likely unaffected in these mutants (Figs 4B and S4A). To address whether loss of *wdr-31* leads to subtle defects in cilia morphology, we expressed the fluorescence-based marker *str-1pro:mCherry*, *gcy-5pro::gfp*, *srb-6pro::gfp*, which label the AWB, ASER, and PHA/PHB cilia, respectively, in *wdr-31* mutants. The AWB dendritic tip extends the fork-like cilia in the WT, and rod-like cilia protrude from the dendritic endings of ASER and PHA/PHB sensory neurons. Our confocal microscopy analysis revealed that the structure and length of ASER, PHA/PHB, and AWB cilia are comparable with those of WT cilia, suggesting, contrary to the importance of WDR31 for ciliogenesis in the zebrafish ear, the loss of *wdr-31* alone does not result in a severe defect in the cilia structure (Fig 4C–G).

The lack of an apparent cilia phenotype in *C. elegans* may be attributed to a functional redundancy for WDR31 in cilia biogenesis, and functional redundancy is indeed a common phenomenon in ciliopathy-related genes. For example, although the absence of individual ciliopathy genes encoding MKS/MKSR proteins or NPHP proteins (NPHP-1, NPHP-4) does not result in serious cilia defect, the loss of *nphp*-4 (the ortholog of human NPHP4) in combination with *mks*-6 (the ortholog of human CC2D2A) causes more severe cilia-related defects (transition zone membrane association defects) (Williams et al, 2011). We next explored the genetic interaction

between *wdr-31* and *elmd-1*, because our results showed that both protein products are at the ciliary base (Fig 1C–F). We discovered that the *wdr-31(tm10423);elmd-1(syb603)* double mutant (hereinafter referred to as "double mutant") has a partial Dyf defect in both head and tail neurons, which was rescued by the expression of a transgene containing the WT *elmd-1* sequence (Figs 5B and S4A and *P* < 0.0001, Fisher's exact test). Using the AWB, ASER, and PHA/PHB cilia fluorescence markers, we subsequently set out to visualize the ciliary structures in the double mutants and single mutants. The AWB cilia in the double mutants have an extra projection in the middle part of the cilia (7% [N: 100] and 52% [N: 134] in WT and double mutants, respectively), whereas ASER, AWB, and PHA/PHB cilia are normal length in WT and double mutants (Figs 4C–F and S4B).

We explored genetic interaction between *wdr-31*, *elmd-1*, and *rpi-2* (the X-linked retinitis pigmentosa protein RP2 and a GAP) because ELMD-1 and WDR-31 localizations are reminiscent of RPI-2 localization (the endogenously labeled RPI-2) in *C. elegans* (Figs 4A and S5A) (Williams et al, 2011). We, therefore, first generated an *elmd-1; rpi-2* double mutant, which revealed no Dyf phenotype (Figs 4B and S4A). However, when we created *wdr-31;elmd-1;rpi-2* triple mutants (hereinafter referred to as "triple mutant"), we observed severe synthetic Dyf phenotype, which is significantly rescued by the introduction of a WT copy of *wdr-31* or *elmd-1* or *rpi-2* (Figs 4B, S3C, and S4A *P* < 0.0001, Fisher's exact test).

We then sought to examine AWB, ASER, and PHA/PHB cilia morphology in triple mutants and compared them with the *wdr-31; elmd-1* and *rpi-2;elmd-1* double mutants. The Dyf defect of triple mutants was indeed accompanied by significant changes in these cilia. The AWB cilia displayed the ectopic projections, including a backward projection from the base of AWB cilia in triple mutant and ectopic projections from the middle part of AWB cilia (0% backward projection in WT vs over 30% backward projection in triple mutants; *P* < 0.0001, Fisher's exact test) (Fig 4F and G). It is noteworthy that the similar backward projections in AWB cilia were observed in two independent triple mutants generated with *wdr-31(syb1568)* and *wdr-31(tur003)* (Fig S6A and B). A backward projection from the ciliary base was observed for ASER cilia, but not PHA/PHB cilia in triple mutants (Fig 4F). We next measured the cilia lengths of these cilia and discovered that ASER and PHA/PHB cilia are significantly shorter in triple mutants (20% shorter for ASER in triple mutants, 13% shorter for PHA/PHB in triple mutants), whereas the cilia length of AWB cilia was not significantly altered in triple mutants (Figs 4C, D, and F and S4B). This establishes a role for WDR-31 in controlling cilia length and morphology in a subset of sensory neurons in a redundant manner with RPI-2 and ELMD-1.

### WDR-31-ELMD-1-RPI-2 are needed for effective cilia entry and exit of IFT components

We next aimed to investigate the mechanism by which WDR-31, ELMD-1, and RPI-2 regulate cilia morphology in *C. elegans*. To this end, we investigated whether these genes affect the localization of IFT proteins and IFT motors, including kinesin motors (Kinesin II and

---

to the BB and PCMC (the periciliary membrane compartment) proximal to the transition zone. Co-labelling of GFP::ELMD-1 with MKS-6:mCherry marker (transition zone) or IFT-140::mCherry in the tail (phasmids) and head (amphid) neurons. Scale bars: 2 μm. **(G)** Shown are representative localizations of WDR-31 and ELMD-1.

A)

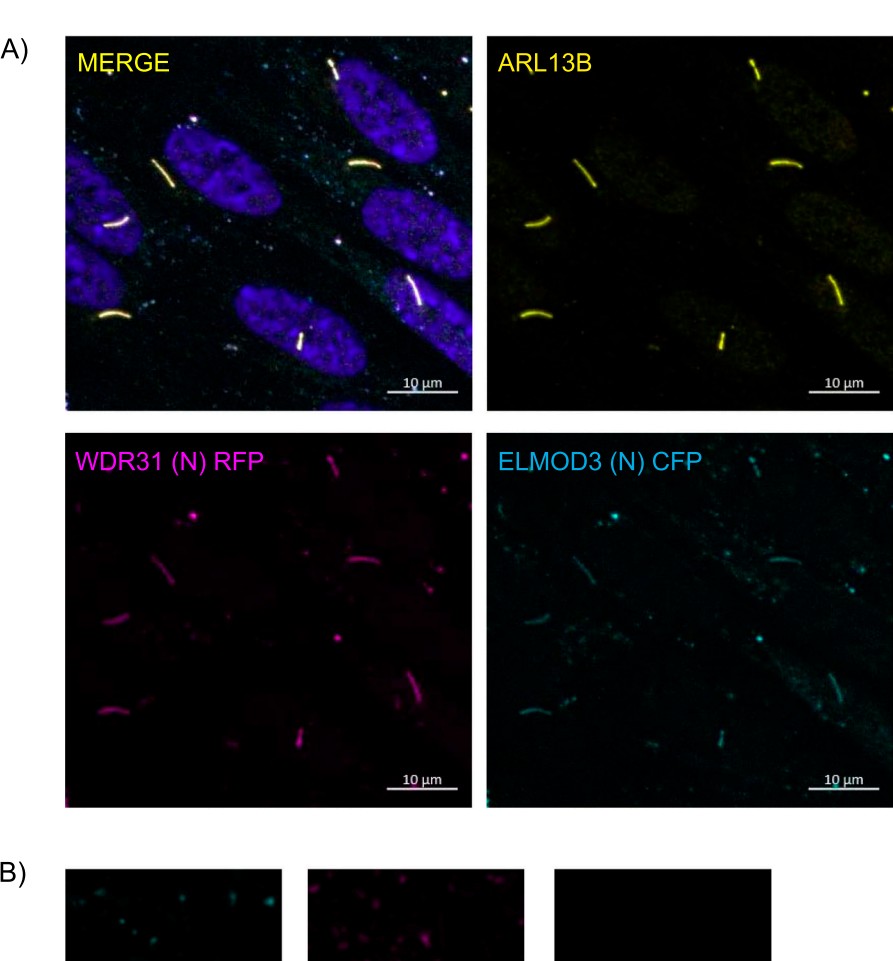

**Figure 2. ELMOD3 and WDR31 localize to the primary cilium.**
**(A, B)** Shown are the staining of WDR31 (tagged with cyan fluorescent protein) and ELMOD3 (tagged with red fluorescent protein) together with a ciliary marker ARL13B and DAPI (nucleus) in hTERT-RPE1 cells. hTERT-RPE1 cells were transiently transfected with 100 ng of WDR31:CFP and ELMOD3:RFP. **(A, B)** Scale bars: 10 μm (A) and 3 μm (B).

B)

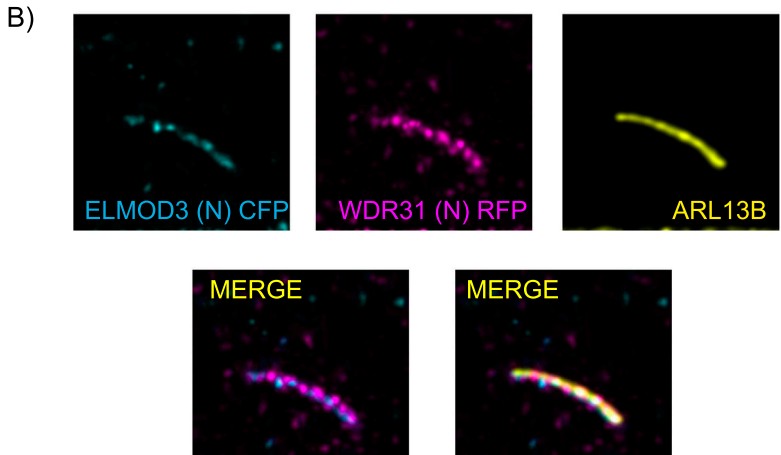

OSM-3/KIF17), dyneins, and IFT components (IFT-A and IFT-B). The IFT complex and motor proteins are critical for cilia construction and maintenance because they deliver ciliary constituents from the cell body to the cilia. In *C. elegans*, two kinesin motors Kinesin II and OSM-3/KIF17 (heterotrimeric kinesin-II and homodimeric OSM-3) work cooperatively to carry the IFT complex in the middle segment of the channel cilia in an anterograde direction, whereas OSM-3/KIF17 transports the entire IFT complex (IFT-A, IFT-B, and BBSome) in the distal part of channel cilia.

We generated single, double, and triple mutants expressing fluorescence-tagged IFT and motor proteins. Using single-copy transgenes, we found that IFT-B components predominantly accumulate at cilia tips and/or middle of cilia in double mutants (96% ciliary tip accumulations for IFT-74/IFT74, N: 26 and 82% ciliary tip accumulations for OSM-6/IFT52, N: 46) but the localization of GFP:: CHE-3 (human dynein heavy chain DYNC2H1) remains less affected in double mutants (27% minor middle cilia accumulations for CHE-3; N: 48, see Video 1) (Figs 5A, C, and D and S8A and Video 1, Video 2, and Video 3 and Table S1). Furthermore, we next investigated the localization of GFP::CHE-3 in triple mutants because the cilia morphology of triple mutants is more severe than that of *wdr-31;elmd-1* or *wdr-31;rpi-2* or *elmd-1;rpi-2* double mutants (Fig 5A). Compared

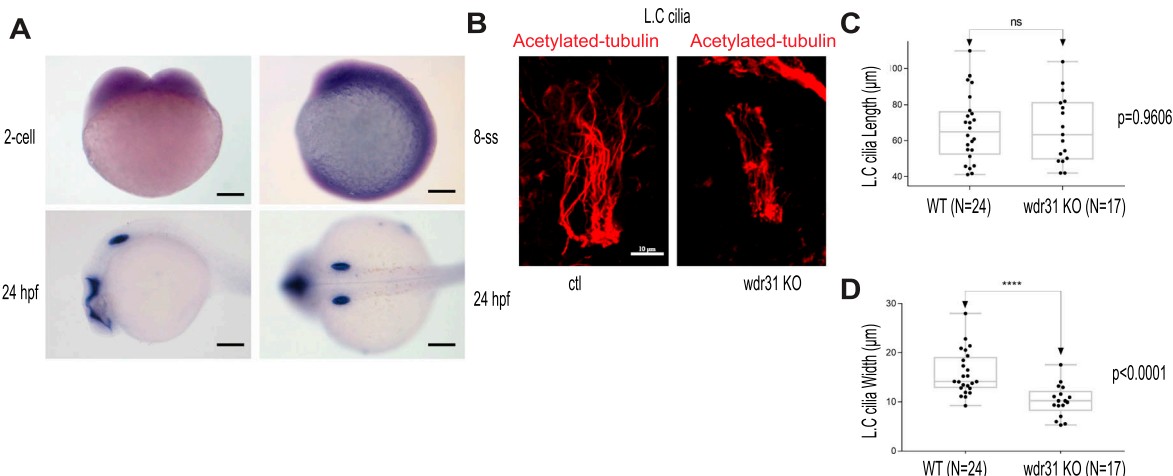

**Figure 3. WDR31 regulates ciliogenesis in zebrafish.**
**(A)** Shown is the expression pattern analysis of *wdr31* in zebrafish embryos. *wdr31* is ubiquitously expressed before the segmentation stage (two-cell and eight-somite stages [SS]). The expression of *wdr31* becomes limited to the otic vesicle and brain region at 24 hours post fertilization (24 hpf). **(B)** Shown are the cilia of lateral crista (LC) of the otic vesicle, stained with acetylated-tubulin, in WT and *wdr31* knockout, generated via CRISPR/Cas9. **(C, D)** The length of cilia in the LC of the otic vesicle remains unaffected in zebrafish *wdr31* knockout, whereas the cilia number is decreased, as shown with the measurement of width of cilia LC of the otic vesicle.

with WT and double mutants, we noticed additional IFT abnormalities in two independent triple mutants, including dim cilia staining with GFP::CHE-3 and accumulations of GFP::CHE-3 in the middle/distal part of cilia (81% middle cilia accumulations, N: 96; 6.6% ciliary tip accumulations, N: 90) (Fig 5A and Video 1). Motor protein OSM-3/KIF17 accumulation within the ciliary tips was detected in double and triple mutants (Fig 5B and Video 4). Furthermore, in triple mutants, the dim distal cilia staining was observed for IFT-140::GFP (IFT-A component) (63% dim distal cilia, N: 47) and XBX-1::mCherry (a dynein light intermediate chain) (Fig 5E and F and Video 5), but the ciliary distribution of KAP-1::GFP (Kinesin II) remains unaffected (Fig 5G).

Cilia accumulations of IFT-B components and OSM-3/KIF17 coupled with weak cilia staining with IFT-A components in triple mutants forced us to better understand the role of WDR-31-ELMD-1-RPI-2 in IFT defects; we therefore employed a in vivo time-lapse video coupled with kymography. For a subset of IFT components, we detected a significant decline in the quantity of IFT particles traveling along the cilia in both directions in double and triple mutants (Fig 6A–H). We found significant decline in the IFT transport for kinesin motor OSM-3::GFP in double and triple mutants in both anterograde and retrograde directions (the anterograde and retrograde: 0.68 n/s and 0.61 n/s in WT; 0.43 n/s and 0.32 n/s in double mutants; 0.34 n/s and 0.23 n/s in triple mutants, *P* < 0.0001; the Mann–Whitney *U* test) (Fig 6A and B), whereas our analysis revealed that the flux of cytoplasmic dynein motor protein CHE-3 along the cilium was unchanged in both directions in both double and triple mutants (data not shown), suggesting, indicating that the simultaneous elimination of these genes has an effect on the kinesin motor OSM-3 but not on cytoplasmic dynein loading onto IFT, but their absence leads to both ciliary cytoplasmic dynein accumulation and the kinesin motor OSM-3 (Fig 5A and B). In WT, the average of IFT-74 particles moving in anterograde and retrograde directions is 0.58 (n/s: number of particles/seconds) and 0.52 n/s,

respectively, whereas in triple mutants, the anterograde and retrograde IFT-74 particles are 0.27 n/s and 0.09 n/s. Comparable reductions were observed for IFT-74::GFP (CRISPR-tagged endogenous IFT-74) in *wdr-31;eldm-1* double mutants (0.36 n/s and 0.21 n/s in *wdr-31;eldm-1; P* < 0.0001; the Mann–Whitney *U* test) (Fig 6G and H). For OSM-6::GFP, the anterograde and retrograde IFT particles statistically differ between WT and triple mutants (the anterograde and retrograde: 0.52 n/s and 0.58 n/s in WT, 0.36 and 0.30 in triple mutants; *P* < 0.0001; the Mann–Whitney *U* test) (Fig 6E and F). Furthermore, IFT-A component IFT-140 transport declined in both directions both double and triple mutants (Fig 6C and D). Taken together, our fluorescent microscopy analysis reveals several IFT abnormalities in triple mutants: first, in triple mutants, there is a significant decrease in the quantity of IFT particles, including the OSM-3/KIF17 kinesin motor and both IFT-A and IFT-B components, traveling in both anterograde and retrograde directions. Second, the ciliary accumulations of IFT-B components and OSM-3/KIF17 kinesin motors in cilia together with weak distal cilia staining of the IFT-A component and dynein (XBX-1) in triple mutants suggest that the defects in the return of IFT from the ciliary tips. These defects might stem from a reduction in the loading of IFT components and dynein onto IFT trains at the ciliary base.

To better investigate the role of WDR-31, ELMD-1, and RPI-2 in IFT, we measured the anterograde (middle and distal segment) and retrograde IFT velocities in WT and triple mutants. When compared with WT IFT speeds, the particle distribution of the IFT velocities (anterograde direction) revealed that more IFT particles (IFT-74::GFP, IFT-140::GFP, OSM-3::GFP, and OSM-6::GFP) travelled faster in the middle segments of triple mutants (Fig 8), implying that the integrity of anterograde IFT components is likely compromised. In contrast, in the triple mutant, the retrograde and distal anterograde IFT velocities are unaltered (Fig S7). Taken together, our findings suggest that with the exception of heterotrimeric Kinesin II, all IFT proteins studied, including dynein

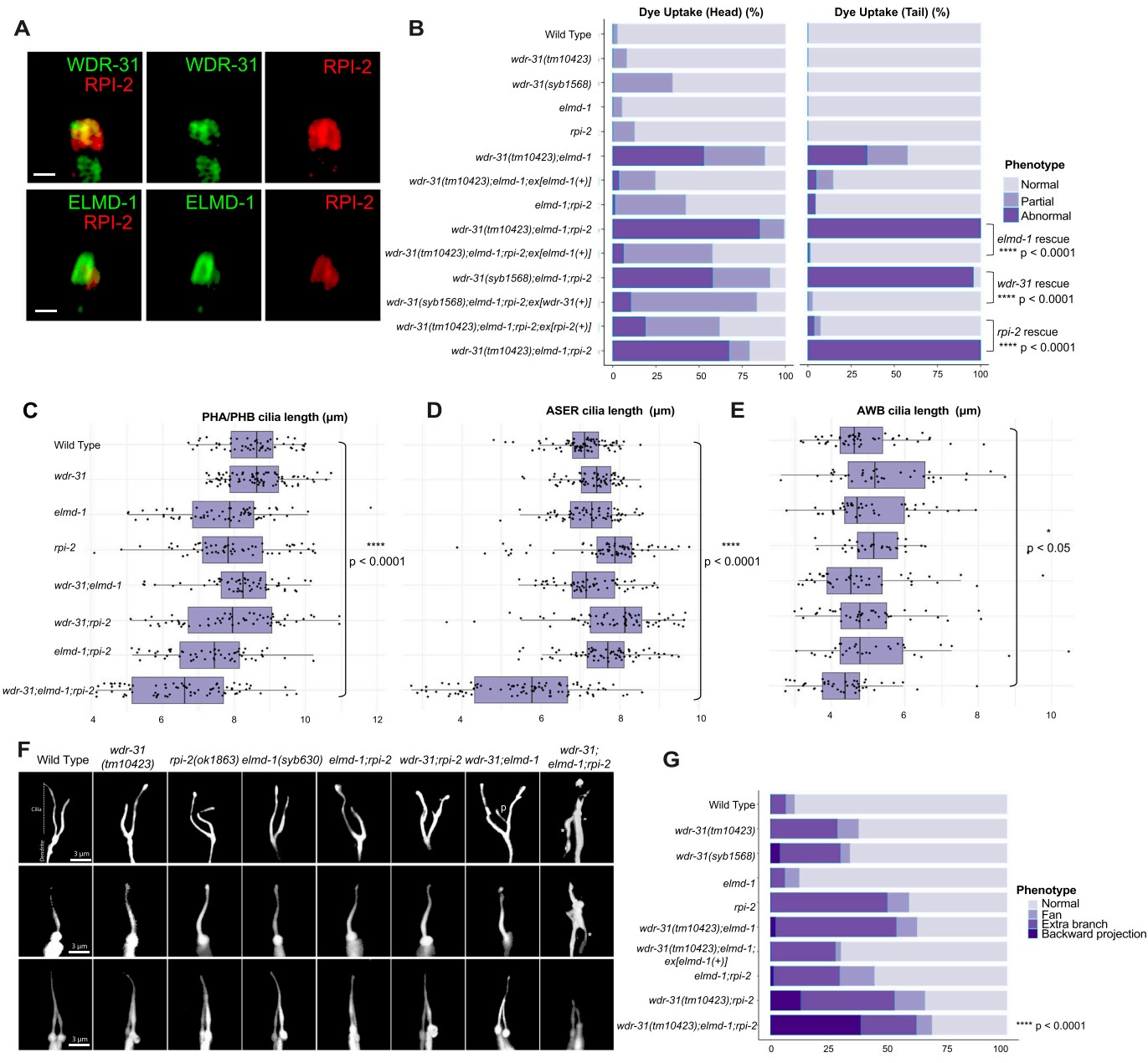

**Figure 4. WDR31-ELMOD-RP2 is needed for determining cilia length and morphology in *C. elegans*.**
**(A)** Shown are co-localizations of RPI-2::mCherry (red, the endogenously labeled RPI-2) with either WDR-31::GFP (green, the endogenously labeled WDR-31) or GFP::ELMD-1 (green, overexpressed) in tail (phasmid) sensory neurons in *C. elegans*. Scale bars: 1 µm. **(B)** The fraction of the dye uptake defects is presented in bar charts for WT and the indicated mutants. Fisher's exact test was performed for statistical analysis between the indicated triple mutants and a rescue gene for Dye assay. Brackets show statistical significance between two strains compared (*P* < 0.0001 and **** indicate statistical significance). **(C, D, E)** Shown is the jitter plot for PHA/PHB cilia, ASER cilia, and AWB short cilia length (µm) for WT and indicated mutant strains. Statistical significance between WT and triple mutants was shown with bracket. **** implies statistical significance, which means that *P* value is lower than *P* < 0.0001, whereas * means that *P*-value is lower than *P* < 0.05. **(F)** Fluorescence images show the morphology of AWB cilia (fork-like structure located in the head), ASER cilia (amphid channel cilia), and PHA/PHB cilia (phasmid channel cilia) in WT and indicated mutant backgrounds. The backward projection from the cilia is shown with asterisks (*), whereas p indicates the ectopic projections from the middle parts of cilia. Scale bars: 3 µm. **(G)** The percentage of the abnormality in AWB cilia morphology is depicted in bar charts. Fisher's exact test was used for statistical analysis of AWB cilia morphology between WT and designated mutants, and **** denotes statistical significance.

CHE-3, OSM-3 kinesin, IFT-B, and IFT-A components, display certain defects in *wdr-31*-related mutants, either ciliary accumulations for dynein CHE-3, OSM-3, and IFT-B component or dim distal cilia staining for IFT-A component and dynein motors (CHE-3 and XBX-1) or changes in IFT velocities.

## WDR-31, ELMD-1, and RPI-2 restrict ciliary entry of non-ciliary proteins

Given that WDR-31, ELMD-1, and RPI-2 localize at the base of cilia, we wanted to examine the role of these proteins in the ciliary gate, and

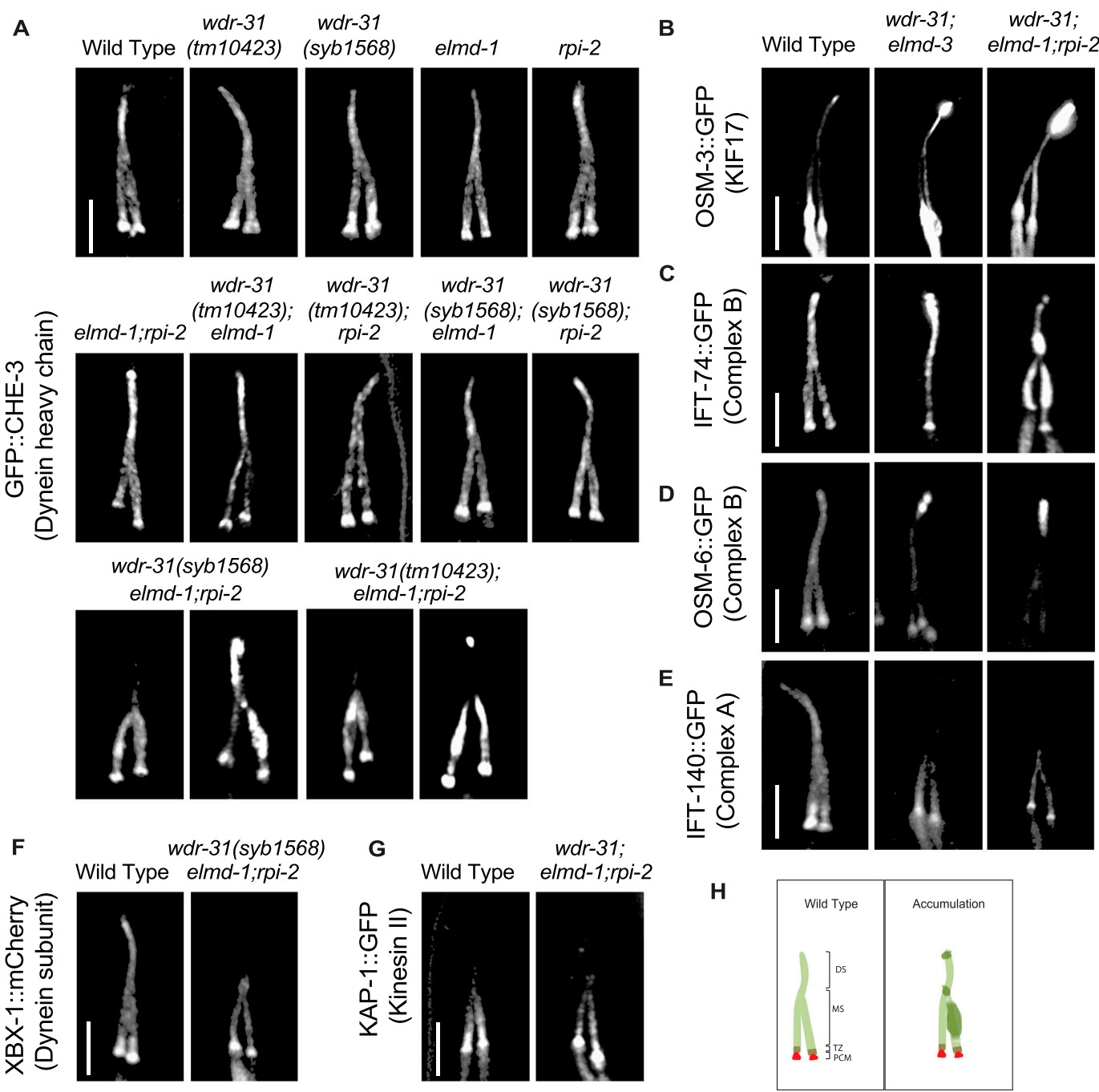

**Figure 5. IFT-B proteins and OSM-3/KIF17 accumulate at the ciliary tip.**
Shown are fluorescent images of PHA/PHB cilia (phasmid tail). **(A)** Fluorescent images from a single copy GFP::CHE-3 (human dynein heavy chain DYNC2H1) in WT and indicated mutant backgrounds are displayed. GFP::CHE-3 accumulations within cilia and dim distal cilia staining were observed in two distinct triple mutants (*wdr-31(tm10423);elmd-1;rpi-2* and *wdr-31(syb1568);elmd-1;rpi-2*). Scale bars: 3 µm. **(B, C, D, E, F)** Confocal microscopy analysis of IFT-A (IFT-140::GFP) and IFT-B complex components (OSM-6/IFT52::GFP and IFT-74::GFP) revealed differential abnormalities in the transport of IFT-A and IFT-B components in double (*wdr-31; elmd-1*) and triple mutants. The localization of XBX-1::mCherry (Dynein subunit) in *wdr-31(syb1568);elmd-1;rpi-2* triple mutants phenocopies the dim distal cilia staining of IFT-A (IFT-140::GFP) in the w*dr-31(tm10423);elmd-1;rpi-2* triple mutants. OSM-3/KIF17 Kinesin motor accumulates at the ciliary tips in w*dr-31(tm10423); elmd-1* double and w*dr-31(tm10423);elmd-1;rpi-2* triple mutants. Compared with double mutants, the ciliary tip staining is stronger in the triple mutants. Scale bars: 3 µm. **(G)** Fluorescent images from Kinesin II motor (KAP-1::GFP) revealed that the restricted middle segment localization of KAP-1 remains unchanged in w*dr-31(tm10423);elmd-1;rpi-2* triple mutants. Scale bars: 3 µm. **(H)** Shown are the drawings of phasmid cilia (PHA/PHB sensory neurons in the tail) in WT and mutants showing ciliary accumulations.

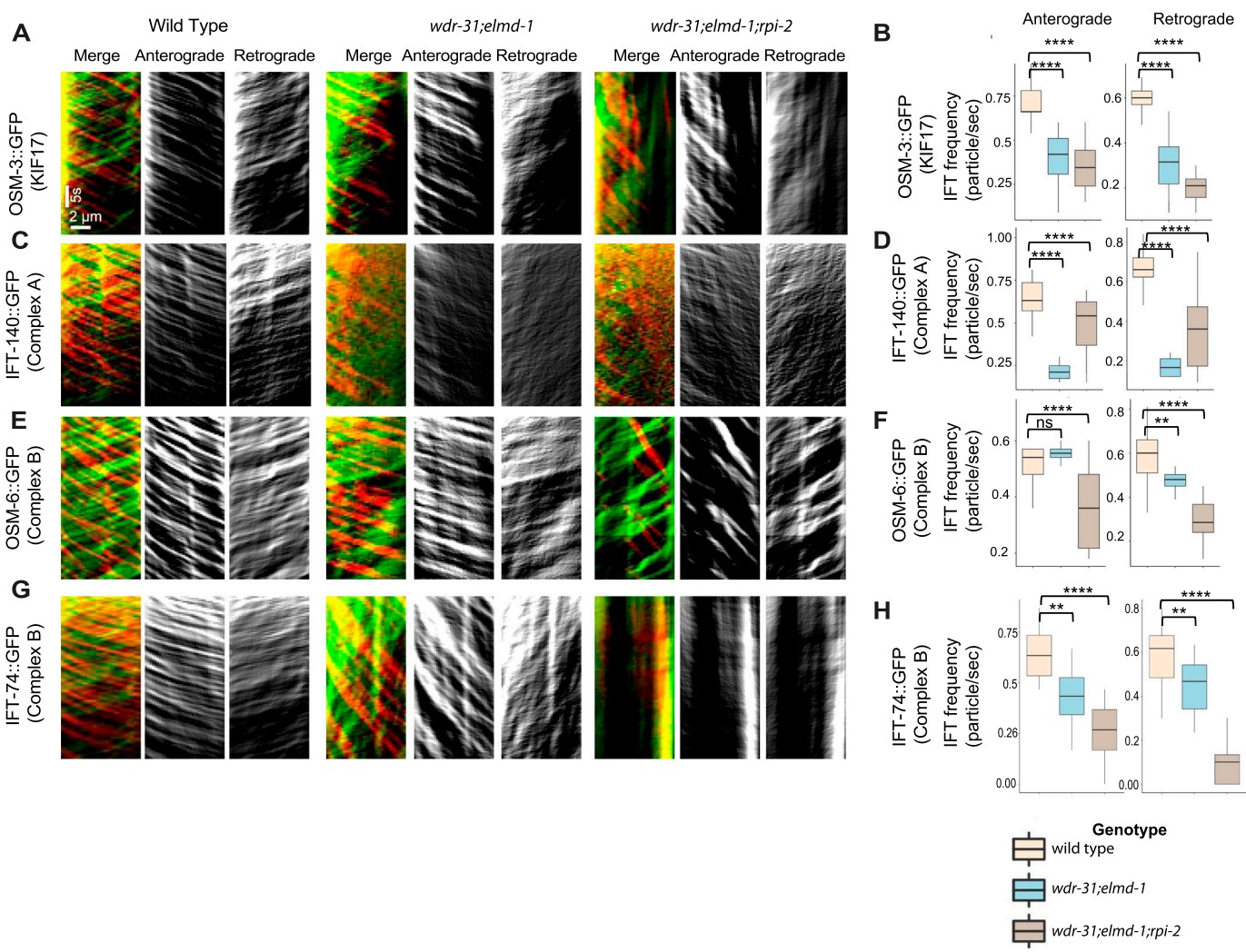

**Figure 6. Measurement of anterograde and retrograde IFT transport frequency.**
**(A, C, E, G)** Shown are representative kymographs of GFP-tagged IFT proteins translocating in the tail cilia (PHA/PHB sensory neurons) of WT and indicated mutants. Kymographs for anterograde, retrogrades, and merged (Red & Green) were generated with ImageJ equipped with KymographClear. The trajectory represents a moving IFT particle, and the average number of moving IFT particles in WT and indicated mutants (double, triple) was calculated by counting all trajectories in each kymograph. Travel time and distance are shown on kymographs. **(B, D, F, H)** Box-and-Whisker charts with error bars were created to visualize the average number of IFT anterograde and retrograde particles between WT and indicated mutants. The Mann–Whitney *U* test was used to measure statistical analysis and significance. The four and three asterisks (**** and ***) at the top of the brackets indicate that the *P*-value between the two strains is less than 0.0001 and 0.001, respectively, suggesting statistical significance. Ns stands for "not significant."

we chose the TRAM-1 protein (the ortholog of human TRAM1), which surrounds the PCMC in *C. elegans* but stays outside of the cilia (Williams et al, 2011), and the transition fiber protein DYF-19 (human FBF1) and the transition zone proteins NPHP-1 (the human NPHP1) and MKS-2 (human TMEM216) (Wei et al, 2013). We showed that neither *wdr-31*, *elmd-1*, or *rpi-2* deletion nor double mutant combinations result in ciliary entry of TRAM-1 or MKS-2 (Fig 7A and B), suggesting that neither of these alone affect ciliary gating. In contrast, TRAM-1 protein leaks into cilia in all three independent *wdr-31* triple mutants, but most of the signal remains outside of the cilia (Fig 7A and B), whereas the localizations of transition zone NPHP-1 and transition fiber protein DYF-19 remain unchanged (Fig 7C). In the *wdr-31;elmd-1;rpi-2* triple mutants, the PLC1-PH::GFP marker for phosphatidylinositol 4,5-bisphosphate (PtdIns(4,5)P2)

stays outside of cilia. In the WT, PLC1-PH::GFP stains PCMC membranes but does not penetrate cilia (Fig 7D). Furthermore, we explored the localization of TAX-4, a ciliary membrane protein, and it remains unaffected in the triple mutants (Fig S8B). Taken together, our findings reveal a functionally redundant role for WDR31, ELMD-1, and RPI-2 in restricting entry of non-ciliary proteins into cilia, despite the fact that TZ protein localization remains unchanged.

**WDR-31 and ELMD-1 are required for recruitment of BBS-7, a BBSome component, to the cilia**

Some of the IFT defects (cilia accumulations of the IFT−B complex, weak cilia staining of IFT-A components, and increase in IFT speed) observed in the *wdr-31;elmd-1;rpi-2* triple mutants are reminiscent

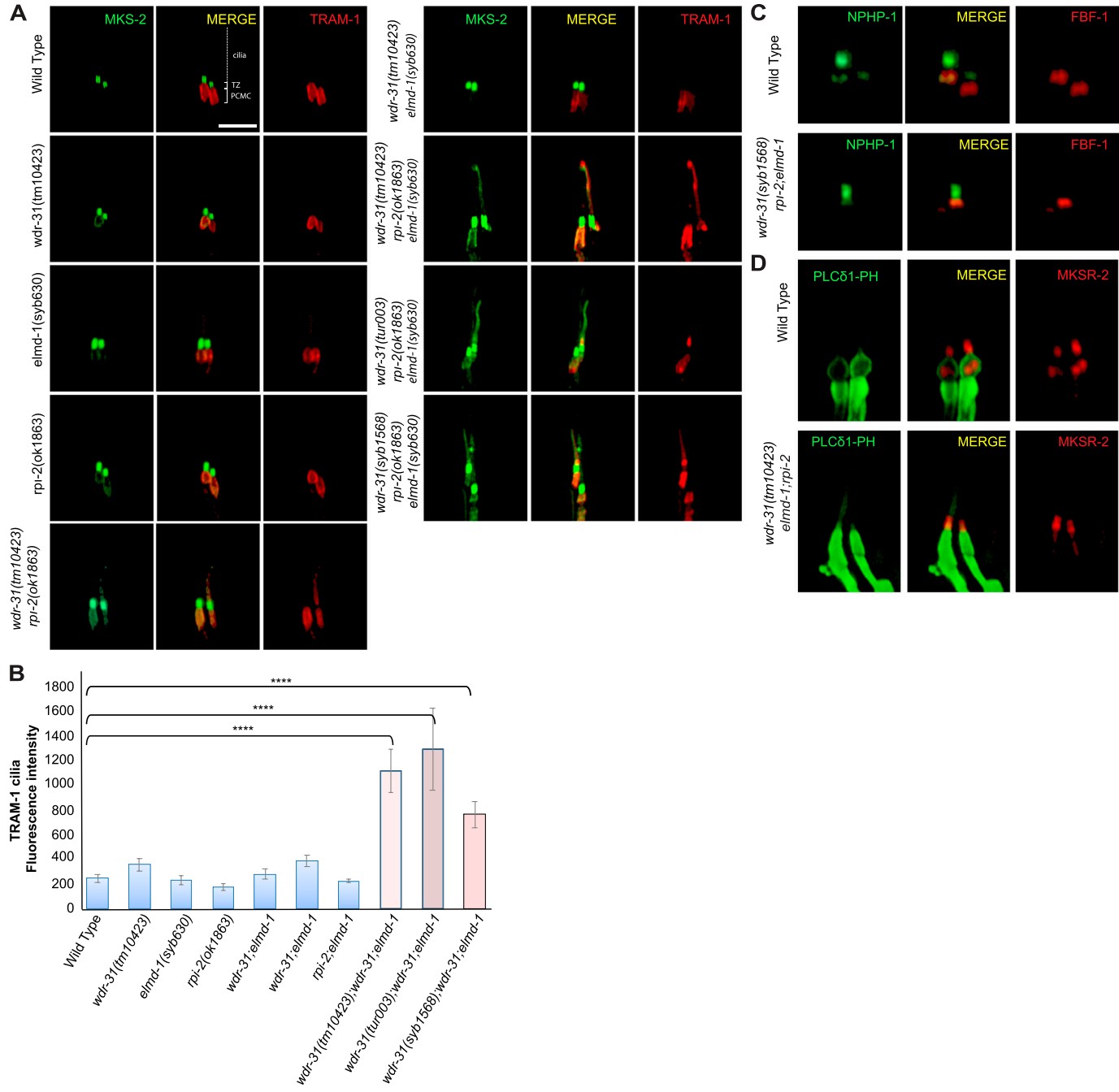

**Figure 7. A non-ciliary membrane protein TRAM-1 enters into the cilia in *wdr-31;elmd-1;rpi-2* triple mutants.**
**(A)** Confocal fluorescent images exhibit the localization of tdTomato-tagged TRAM-1 (a PCMC marker) and MSK-2::GFP (a TZ marker) in WT and indicated mutants. TRAM-1 leaks into the cilia in all three *wdr-31* triple mutants. Cilia, the periciliary membrane compartment (PCMC), and transition zone (TZ) are depicted in the fluorescent image. Scale bars: 2 μm, **(B)** TRAM-1 fluorescence intensities in the cilia were measured in WT and designated mutants, and the results were shown in the plot. The four asterisks (****) indicate statistically significant differences between the WT and the identified triple mutants. **(C, D)** The localization of NPHP-1::GFP (a transition zone protein) and FBF-1::mCherry (a transition fiber protein) was similar unaffected in *wdr-31;elmd-1;rpi-2* triple mutants **(D)** The PLCδ1-PH::GFP (a marker for monitoring phosphatidylinositol 4,5-bisphosphate (PtdIns(4,5)P2) in the plasma membrane) decorates the membranes of PCMC and does not enter into the cilia in WT. MKSR-2 was used to mark the transition zone. The PLCδ1-PH::GFP stays outside of the cilia in the *wdr-31;elmd-1;rpi-2* triple mutants.

of IFT defects observed in several *bbs* mutants (Blacque, 2004; Ou et al, 2005; Wei et al, 2012; Xu et al, 2015). We hypothesized that the IFT complex destabilization defects in triple mutants might be because of defects in the BBSome. First, we found that ciliary tip accumulations of IFT-74:GFP, an IFT-B component, in *bbs-8(nx77)* mutants were comparably similar to that of double and triple mutants (Fig 9A). Furthermore, there is no further increase of IFT accumulations in the triple mutants *wdr-31;elmd-1;bbs-8*.

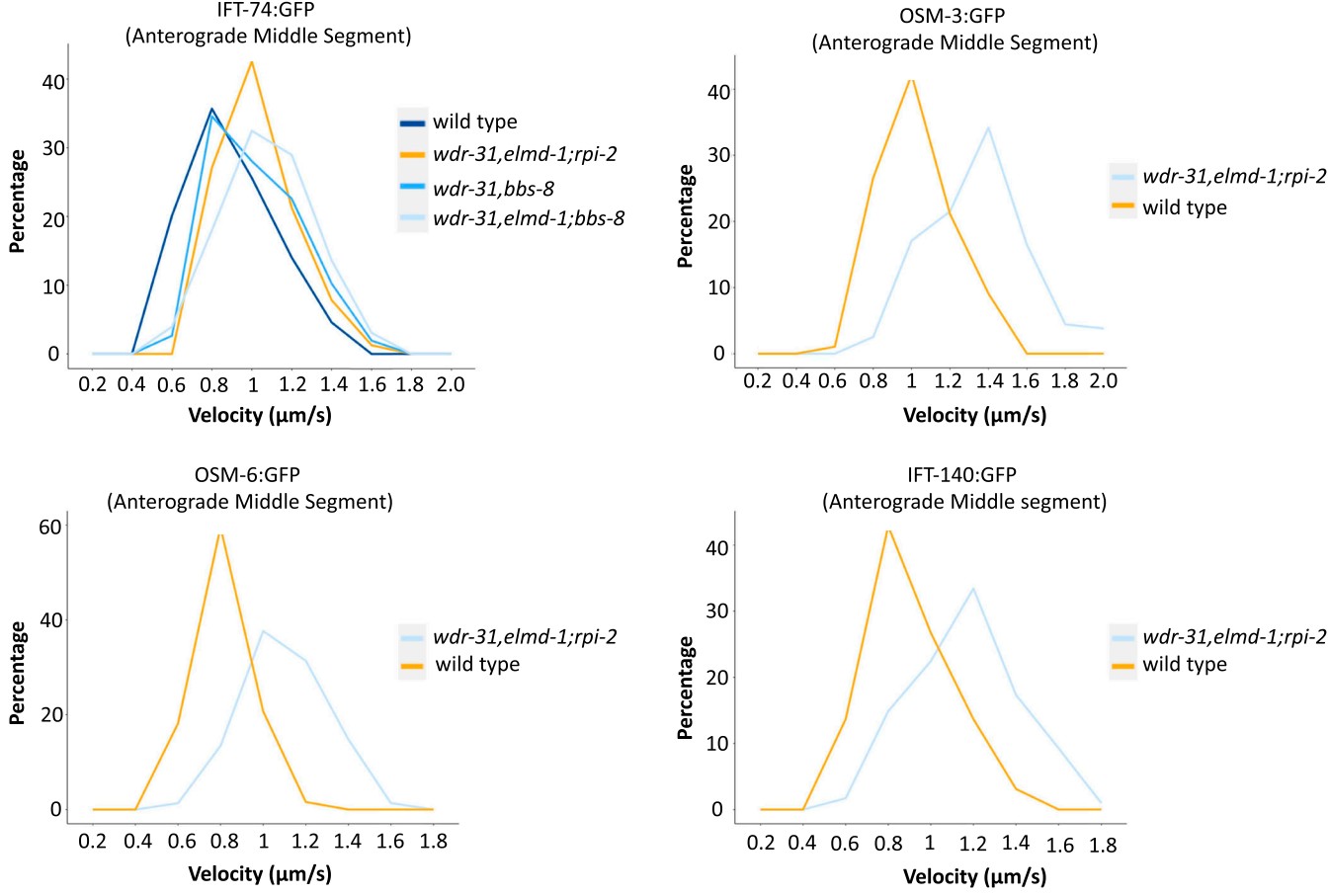

**Figure 8.  The anterograde IFT velocities switched in favor of faster speeds in _wdr-13;elmd-1;rpi-2_ triple mutants.**
The percentage distribution (particles) of the indicated IFT velocities (anterograde middle segment) in the WT and corresponding mutants is shown in the velocity versus percentage line graphs. The complete list of proteins that interact with WDR31 and ELMOD3 is provided in Table S2.

We then investigated whether the ciliary localization of BBS-7:: GFP, a core member of the BBSome, is controlled by WDR-31, ELMD-1, and RPI-2, because the ciliary localization of other BBSome components is dependent on other BBSome subunits (Ou et al, 2007). We predict that the ciliary localization of BBSome is more likely disturbed in triple mutants, contributing to the observed IFT abnormalities. Consistent with expectations, BBS-7::GFP is lost or significantly diminished in the cilia of double and triple mutants, but BBS-7::GFP localization persists at the ciliary base in these mutants (Fig 9B). We next performed the time-lapse movie analysis of BBS-7::GFP with kymography, which revealed that BBS-7::GFP movements were undetectable in 27% and 42% of cilia in the head in double and triple mutants, respectively (Video 6). We were able to quantify the frequency of BBS-7::GFP even though the density of trajectories (IFT particles) on kymographs was weak in double and triple mutants relative to WT (Fig 9C). Our analysis revealed a substantial decrease in the average of BBS-7::GFP particles translocating in the anterograde and retrograde directions in the remaining worms that displayed IFT (average antero-grade and retrograde IFT particles: 0.55 n/s and 0.58 for WT; 0.30 and 0.33 n/s for double mutants; 0.07 and 0.07 n/s for triple mutants, P < 0.0001; the Mann–Whitney U test) (Fig 9C and D and Video 6). We

next used IFT-74::GFP to investigate IFT speeds in _bbs-8_ and its mutant combinations with _wdr-31_ and _elmd-1_. The velocity distri-bution of IFT-74::GFP particles in the middle segment revealed a comparable resemblance between the _bbs-8;wdr-31;elmd-1_ triple mutant and the _wdr-31;elmd-1;rpi-2 t_riple mutant, with more par-ticles shifting toward fast (Fig 8). Taken together, our results suggest that because the BBSome complex is unable to gain access to the cilia in double and triple mutants, IFT-B components, and OSM-3/ KIF17 likely accumulate excessively in the distal portion of the cilia, whereas IFT Complex A component staining the distal segment cilia becomes dim (Fig 9E).

# Discussion

### WDR31 is a new ciliary protein encoding gene required for cilia biogenesis

Clinical genomics, proteomics, functional genomics, and bio-informatics research have all contributed to the expansion of the list of both ciliary and ciliopathy genes. Over 300 genes have been

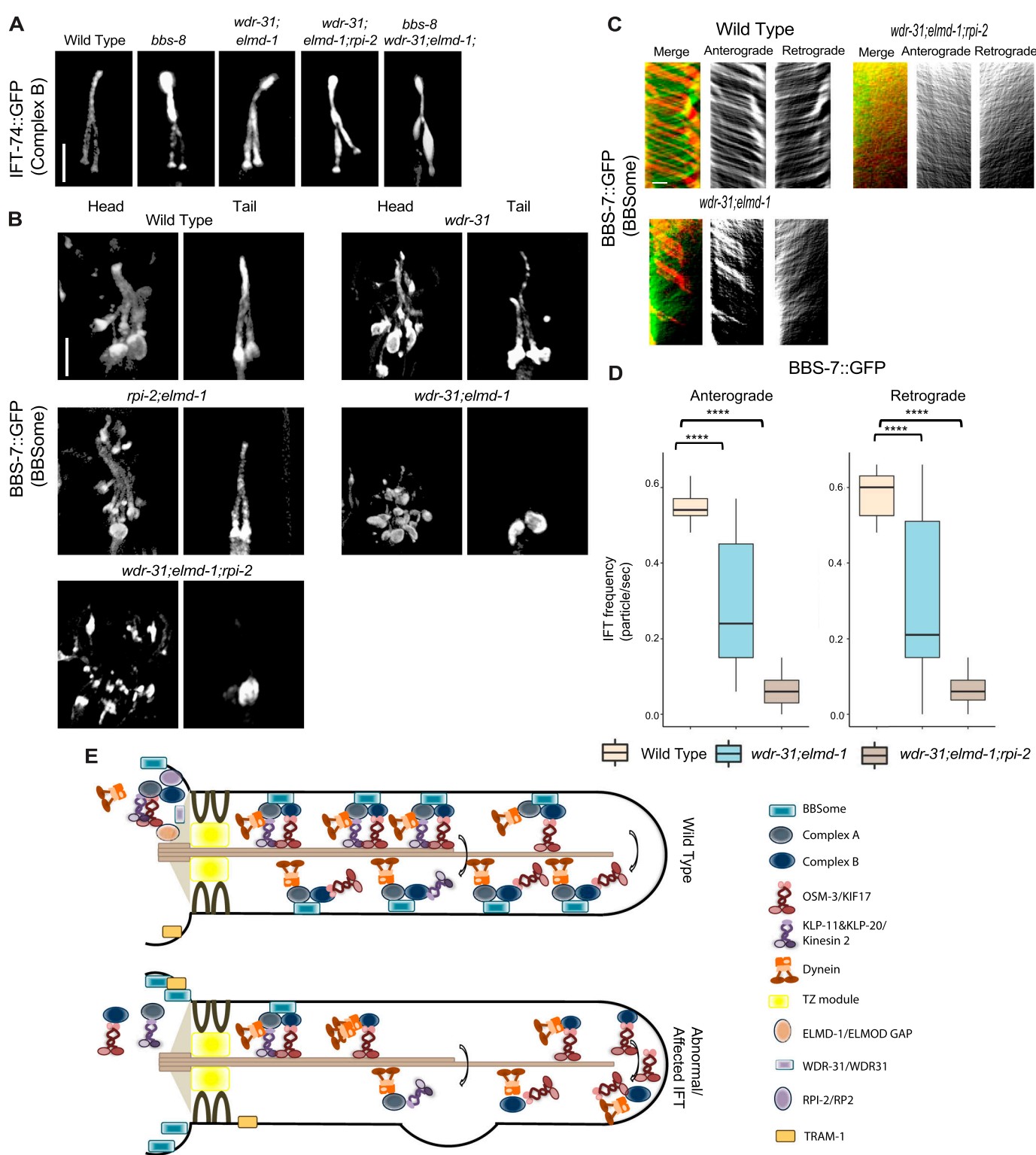

**Figure 9. WDR-31 and ELMD-1 regulate the recruitment of BBSome to the cilia.**

**(A)** Shown are fluorescence images from the transgenic strain carrying IFT-74::GFP, an IFT-B component, in WT, *wdr-31;elmd-1* double mutants, *wdr-31;elmd-1;rpi-2* and *wdr-31;elmd-1;bbs-8* triple mutants, and *bbs-8(nx77)*. The IFT-B subunit IFT-74:GFP accumulates at the ciliary tips and cilia in the tails of all three mutants. **(B)** Confocal fluorescence images showing the localization of BBS-7:GFP, a BBSome subunit, in the heads and tails of WT and *wdr-31(tm10423); wdr-31;elmd-1*; *rpi-2;elmd-1* double and *wdr-31;elmd-1;rpi-2* triple mutants. Fluorescence images showed absent or weak cilia staining of BBS-7::GFP in both the heads and tails of *wdr-31;elmd-1* and *wdr-31; elmd-1;rpi-2* triple mutants. **(C)** Kymographs were created from time-lapse BBS-7::GFP movies (PHA/PHB cilia) using KymographClear integrated into ImageJ. Shown are representative kymographs for BBS-7::GFP translocating in WT and indicated mutants. Each trajectory in kymographs was counted. **(D)** Travel time and distance are

identified as encoding ciliary proteins, including IFT-kinesin-dynein components, BSSome components, structural ciliary components, signaling molecules, and IFT regulators, with many of them being ciliopathy genes (Genomics England Research Consortium et al, 2019; Vasquez et al, 2021). Our findings reveal the roles of WDR-31, a member of the WD40-repeat protein (WDR) family, the ELMOD orthologue (ELMD-1), and RP2 orthologue (retinitis pigmentosa 2; RPI-2) in cilia and IFT regulation. Several members of the WD40-repeat protein family, including WDR34, WDR35, and WDR60, have previously been connected to the cilia, but the role of *WDR-31* in cilia has been unknown (Blacque et al, 2006; Rompolas et al, 2007; Patel-King et al, 2013). Strikingly, we found that WDR31 is a ciliary protein, and knocking it out in zebrafish using CRISPR/Cas9 causes ciliary defects, including a decrease in the number of cilia in the LC. Furthermore, the link between WDR31 and cilia was significantly strengthened by findings from *C. elegans*. First, the expression of the *WDR31* orthologue (WDR-31) is restricted to the ciliated sensory neurons, and WDR-31 is localized to ciliary compartments (the ciliary base and cilia) in *C. elegans* and human TERT-RPE1 cells. The exclusive cilia expression pattern of *wdr-31* in *C. elegans* is similar to that of ciliary and ciliopathy genes like ARL-13/ALR13B, IFT, and BBS (Blacque et al, 2005). Second, CRISPR/Cas9-mediated knock out of *WDR-31* combination with the ELMOD orthologue (ELMD-1) and RP2 orthologue (retinitis pigmentosa 2; RPI-2) results in an abnormality in cilia functions and structures in *C. elegans*. Our findings reveal that WDR31 is a new ciliary protein required for cilia morphology in both zebrafish and *C. elegans*, suggesting the evolutionarily conserved role of WDR31 in cilia biology.

## Functional redundancy between WDR-31, ELMD-1, and RPI-2

Our genetic analysis in *C. elegans* provides evidence for functional redundancy for *wdr-31*, *elmd-1*, and *rpi-2* in regulating cilia length, cilia morphology, and the trafficking of kinesin-IFT–BBSome complexes. Though cilia length and cilia morphology are disrupted in *wdr-31;elmd-1;rpi-2* triple mutants, with altered recruitment of kinesin, IFT, and BBS proteins and IFT dynamics, simultaneous elimination of *elmd-1* and *rpi-2* did not result in severe defects in the cilia and IFT. Our interpretation for these data is that the additive defects in the triple mutants were likely because of *wdr-31*, and WDR-31 is a central player in controlling cilia morphology and IFT machinery. However, we cannot rule out the possibility of microtubule defects in cilia ultrastructure in *elmd-1;rpi-2* double mutants. Additional work is needed, and a transmission electron microscope may be used to reveal further details in the ultrastructure of cilia in these double mutants. Furthermore, it is interesting to note that *wdr-31* mutations in *C. elegans* and zebrafish have distinct ciliary abnormalities; this might be because WDR-31

functions redundantly with ELMD-1 and RPI-2 in *C. elegans*. It would be helpful to knock out WDR31 in mammals to comprehend the significance of WDR31 in mammals.

How do these proteins work together to regulate cilia-related phenotypes? Although WDR31 is a poorly characterized gene, the roles of human RP2 and ELMOD proteins in cilia biology are better understood. Human RP2 localizes to the cilium and the basal body and displays GAP activity toward two ARF family members, ARL2 and ARL3, both of which were linked to cilia biology (Evans et al, 2010; Wright et al, 2011; Schwarz et al, 2017), whereas the ELMOD protein family (ELMOD1-3) has GAP activity for ARL2 (ELMOD1 and ELMOD3) and ARF6 (ELMOD1) (Johnson et al, 2012; Jaworek et al, 2013; Ivanova et al, 2014; Miryounesi et al, 2019). ELMOD2 localizes to the basal body, and our study showed ELMOD3 localizes to cilium (Turn et al, 2021). Taking into account the fact that RPI-2 and ELMD-1 are GAPs, and loss of these two GAP proteins likely result in the overactivation of their target G proteins, how does WDR-31 function with these GAP proteins? One possibility is that WDR31 may function downstream of regulatory GTPases, including ARL2, ARL3 or unidentified GTPases, activated by these GAP proteins. One downside of this explanation is that *wdr-31* single mutants did not have significant ciliary and IFT defects; but, if this were valid, we would expect to observe further anomalies in cilia and IFT in *wdr-31* single mutants, close to the removal of these three genes. Alternatively, the activity/function of overactive GTPases can be somehow regulated by WDR-31, maybe WDR-31 may have a GAP activity for the regulatory GTPases or it may have a GTPase activity. Although WDR-31 does not seem to have a GAP domain, HHMER search revealed that human WDR31 has a distant sequence resemblance to a nucleoside-triphosphatase (NTPase) domain (NACHT) containing protein in *Penicillium camemberti* (Gabler et al, 2020). PSI-BLAST search confirmed this result (unpublished data).

Furthermore, the ciliary roles of these two GAP proteins might be independent of their GAP activities, and consistent with this idea, the recent study showed the ciliary roles of ELMOD2 are partially independent of its GAP activity (Turn et al, 2021). However, a variety of evidence suggests that WDR31 and ELMOD3 do not form a complex. First, mass spectrometry-based proteomic analysis for either WDR31 or ELMOD3 showed that neither WDR31 nor ELMOD3 contains each other or RP2, but we cannot completely rule out a transient and temporary encounter between them (Table S2 and Fig S9A and B). Second, the proper localization of any of these three proteins was independent of each other (unpublished data). Further studies are needed to understand a mechanistic link between WDR31, ELMOD, and RP2, and figure out the independent contributions of WDR31 and both GAP proteins in cilia biology.

included on kymograph (D) The graph depicts the average number of BBS-7::GFP particles traveling around the cilia in both directions for WT and indicated mutants. **(E)** The Mann–Whitney *U* test revealed statistical significance between the compared strains and that the *P*-value was less than 0.0001 shown by the four asterisks (****) at the top of the brackets (E) In WT, the assembly of the Kinesin–IFT-−BBSome complex (Kinesin-II and OSM-3, IFT-B, IFT-A, and BBSome) happens at the base of the cilia. In the middle segment of amphid and phasmid cilia in *C. elegans*, both heterotrimeric Kinesin II and homodimeric OSM-3 transport the IFT-A–IFT-B–BBSome complex in an anterograde direction. Heterotrimeric Kinesin II returns to the ciliary base when it reaches the tip of the middle segment of amphid and phasmid cilia, whereas homodimeric OSM-3 is responsible for the anterograde translocation of the IFT–BBSome complex in the distal segment of amphid and phasmid cilia. When the OSM-3–IFT–BBSome complex reaches the ciliary tip, cytoplasmic dynein transports them back to the ciliary base. In *wdr-31;elmd-1* double *wdr-31;elmd-1;rpi-2* triple mutants, the BBSome failed to enter into the cilia, thus leading to accumulations of OSM-3 and IFT-B components in the ciliary tips.

## A potential function for WDR31–ELMOD–RP2 in determining the integrity of ciliary gate

Our findings that worm lacking WDR31–ELMOD–RP2 also display ciliary entrance of non-ciliary proteins (TRAM-1) suggests that these proteins may play a function in the ciliary gate. Despite the mechanism by which these proteins regulate the ciliary gate functions remains unknown, the abnormality of IFT proteins (abnormal distribution and increased anterograde IFT velocities) in triple mutants may provide some hint about the potential roles of these proteins in the ciliary gate. One possible explanation is that the IFT defect in triple mutants potentially results in compromised ciliary gate function. Consistent with this possibility, it was previously reported that in *C. elegans*, the elimination of IFT complex A components and cytoplasmic dynein motor CHE-3 result in a defect in ciliary gate function (Jensen et al, 2018; Scheidel & Blacque, 2018).

## WDR-31–ELMOD–RP2 is needed for efficient IFT entry and IFT dynamics

Our findings show that simultaneous disruption of *WDR-31*, *ELMD-1*, and *RPI-2* causes differential effects on ciliary IFT protein localizations, including accumulation of IFT-B subcomplex components (IFT-74 and OSM-6), dynein (CHE-3), and kinesin motor (OSM-3) or dim cilia staining of IFT-A subcomplex component and a dynein component (CHE-11 and XBX-1) or almost no cilia entry of a BBSome component (BBS-7). Furthermore, with the exception of cytoplasmic dynein CHE-3, the triple mutant shows a significant decrease in moving IFT particles along cilia in both directions for almost all IFT proteins, indicating that the combined elimination of WDR-31, ELMD-1, and RPI-2 has a significant impact on ciliary recruitment of IFT/BBSome components (IFT-74, OSM-6, CHE-11, and BBS-7) and OSM-3. Our in vivo time-lapse movie analysis indicated that anterograde IFT velocities for OSM-3, OSM-6, IFT-74, and CHE-11 in the middle segment of cilia are considerably increased in the triple mutant, whereas IFT velocities in the distal segment of cilia and retrograde direction are unaltered (Fig 8A–D). This suggests that in cilia lacking WDR-31, ELMD-1 or RPI-2, certain portions of these IFT proteins are not effectively loaded onto anterograde IFT.

Based on our results, which involve an increase in anterograde IFT speed, a decrease in moving IFT particles in anterograde and retrograde transport, ciliary tip aggregation of OSM-3/KIF17 and IFT-B, and failure of docking of BBSome to the IFT machinery, we propose that some of these defects likely stem from the failure of BBSome to associate with moving IFT in the cilia in double and triple mutants. These proteins are likely regulator for IFT/BBSome. Consistent with our proposal, the almost comparable IFT/BBSome phenotypes were reported with hypomorphic mutations in *dyf-2* (WDR19 orthologue, an IFT-A component) and *bbs-1* (a BBSome component), where they showed that although BBSome remains at the base of cilia and does not undergo IFT, the failure of IFT-B reassociation with IFT-A at the ciliary tips results in IFT-B ciliary tip accumulations, despite the fact the association of IFT-A and IFT-B components in anterograde transport persists (Wei et al, 2012). They proposed that the BBSome controls the IFT assembly at the ciliary base and IFT turnover at the ciliary tip. However, several of the abnormalities identified in the triple mutant, such as diminished

IFT frequencies and ectopic projection from the ciliary base, are unlikely to be related to BBSome dysfunction in the triple mutant, because our findings revealed that the IFT frequency in the *bbs-8* mutant stays intact. Plus, we did not observe an ectopic project from the ciliary base in the *bbs-8* mutant (data not shown).

However, the IFT defects in the triple mutant are unlikely to be because of mislocalization of transition fiber proteins such as DYF-19 (the FBF1 orthologue), which was implicated in regulating the ciliary entry of IFT and BBSome, because our data showed that the localization of DYF-19 was not impaired in the triple mutants (Wei et al, 2013).

On the other hand, the effect on IFT might be direct because RP2 was previously proposed to control the IFT protein IFT20 pool in the peri-basal body and trafficking of Kif17 and Kif7 to the ciliary tip (Schwarz et al, 2017). However, several interesting questions have yet to be resolved: what is the mechanism by which WDR-31–ELMD-1–RPI-2 regulates the ciliary entry of the BBSome entry? How does WDR-31-ELMD-1-RPI-2 interact with proteins, including ARL6, that regulate the BBSome recruitment (Jin et al, 2010)? Future study research will help us to understand how these proteins regulate BBSome/IFT trafficking. In summary, this study provides a mechanistic explanation of the function of WDR-31, a new ciliary protein, ELMD-1, and RPI-2 in the regulation of cilia biogenesis and also contributes a new regulator for IFT/BBSome.

# Materials and Methods

### *C. elegans* strains, maintenance, and genetic crossing

For strain maintenance and genetic crosses, a standard procedure was followed, as described by Sidney Brenner in 1974 (Brenner, 1974). After genetic cross with a marker to generate single, double, and triple mutants, we used the PCR strategy to trace the mutations in the following mutants: *wdr-31(T05A8.5)(tm10423)II.; wdr-31(tur003)II.; wdr-31(syb1568)II.; nphp-4(tm925)V.; mks-6(gk674)I.; elmd-1(syb630) III.; RB1550 rpi-2(K08D12.2)(ok1863) IV.;* and *bbs-8(nx77) V*. Primers can be found in Table S3.

### Lipophilic fluorescent Dye-uptake assay and rescue analysis

Healthy mixed-staged animals were collected using M9 buffer (3 g/l KH2PO4, 6 g/l Na2HPO4, 5 g/l NaCl, 1 mM MgSO4), centrifuged for 1 min at 2,000 rpm, and washed twice with M9 buffer to remove any bacterial contaminations. Worms were incubated for 45–60 min at room temperature in an M9 buffer containing a lipophilic dye (1:200 dilution in M9, Invitrogen Vybrant DiI Cell-Labeling Solution) (Herman & Hedgecock, 1990). The worms were then washed twice with M9 before being moved to a new NGM plate. WT was always included in dye-filling assay, and the dye uptake control for the WT was performed under a stereotype fluorescence microscope, followed by imaging with the fluorescence upright microscope. For the rescue experiment, *N2;turEx24[arl-13p::GFP::elmd-1 (C56G7.3)::unc-54 3'UTR +rol-6}* (OIK1045) were crossed into *wdr-31(T05A8.5)(tm10423)II., elmd-1(syb630) II.,* double and *wdr-31(T05A8.5)(tm10423)II., elmd-1(syb630) II, rpi-2(K08D12.2)(ok1863) IV*

triple mutants. *N2;turEx21[arl-13p::wdr-31 (T05A8.5)::GFP::unc-54 3'UTR +rol-6} (1 ng)* (OIK1042) were mated with *T05A8.5(syb1568) II., elmd-1(syb630) II, rpi-2(K08D12.2)(ok1863) IV* triple mutants. Plasmid (*arl-13p::wdr-31 (T05A8.5)::GFP::unc-54 3'UTR*) was directly microinjected into triple mutant *T05A8.5(syb1568)II., elmd-1(syb630) II, rpi-2(K08D12.2)(ok1863) IV* (1 ng). *dpy-5(e907);nxEx386[rpi-2::gfp + dpy-5(+)]* (MX352) were crossed into *wdr-31(T05A8.5)(tm10423) II., elmd-1(syb630) II, rpi-2(K08D12.2)(ok1863) IV.* triple mutants. Three independent Dye-uptake assays were performed, and fluorescence filters were set for GFP and Texas Red, followed by fluorescence imaging (20–150 heads and tails were counted). Dye uptake of mutants with the corresponding transgenics strains was compared with that of non-transgenic strains in the same rescue plates (Fig S2C and D).

### Generation of mutants using CRISPR/Cas9 in the nematode *Caenorhabditis elegans*

To generate *wdr-31(T05A8.5)(tur003)* allele, three sgRNAs targeting *C. elegans* T05A8.5 (human WDR-31) were chosen using an online tool, Benchling (Biology Software) (2019), followed by ordering complementary oligonucleotides (Macrogen) and cloning of sgRNAs into an empty sgRNA vector pRB1017. The successful sgRNA insert was confirmed with colony PCR, followed by plasmid isolation. Three sgRNAs (each 50 ng/μl) were injected into the gonads of WT together with pDD162 (*Peft-3::Cas9;* 15 ng/μl plasmid pRF4) and pRF4 (50 ng/μl plasmid pRF4) (Dickinson et al, 2013). F1s with the roller phenotype were identified, and after they generated enough progenies, the PCR technique was used to identify the F1 generation bearing the predicted deletion, and the homozygosity of the allele mutation. The PCR products from knockout animals were then sent to the Sanger sequencing (Macrogen). *wdr-31*(tur003)*II.* mutants have 1,276-bp deletion covering a huge part of exon II (297 bp out of 359 bp) and whole exons III, IV, and exon V. This is likely a null allele of *wdr-31*. sgRNA sequences can be found in Table S3.

### Generation of transgenic strains and strain list for *C. elegans*

To generate transgenic lines for localization, rescue experiments, and expression patterns, we generated the following transgenic animals via microinjections.

OIK1042 *turEx21[arl-13p::wdr-31 (T05A8.5)::GFP::unc-54 3'UTR +rol-6}; T05A8.5(syb1568)II., elmd-1(syb630) II, rpi-2(K08D12.2)(ok1863) IV.* (1 ng).

OIK1044 *N2;turEx23[elmd-1p::GFP::elmd-1 (C56G7.3)::unc-54 3'UTR +rol-6}* (5 ng).

OIK1045 *N2;turEx24[arl-13p::GFP::elmd-1 (C56G7.3)::unc-54 3'UTR + rol-6}* (5 ng).

OIK1046 *N2;turEx25[elmd-1p(C56G7.3)::GFP::unc-54 3'UTR +rol-6}* (50 ng).

OIK1047 *N2;turEx26[wdr-31 (T05A8.5)p::GFP::unc-54 3'UTR +rol-6}* (50 ng).

The rol-6 plasmid (50 ng/μl plasmid pRF4) was co-injected as the co-transformation marker. In brief, the plasmids were delivered by microinjections into the gonads of 1-d adult worms. Worms were initially transferred onto a 2.5% agarose pad before (Halocarbon oil, 9002-83-9; Sigma-Aldrich), followed by microinjection. The

microinjection was done using a Zeiss Axio Vert.A1 inverted microscope with DIC optics coupled with a Narishige Micromanipulator MMO-4. We next manually inspected the plates to find successful transgenic animals.

### WT and mutant alleles

N2; FX30333; *wdr-31(T05A8.5)(tm10423)II.;* OIK393 *T05A8.5(tur003)II.;* PHX1568 *T05A8.5(syb1568)II.;* *nphp-4(tm925)V.;* *mks-6(gk674)I.;* PHX630, *elmd-1(syb630) III.;* RB1550 *rpi-2(K08D12.2)(ok1863) IV.;* MX52, *bbs-8(n x 77) V.* We obtained *wdr-31(T05A8.5)(tm10423)II.* (FX30333) mutant allele, which has a 160-bp deletion, causing a frameshift, from the National Bioresource Project. The *wdr-31(T05A8.5)(tm10423) II.* was outcrossed to WT four times. The Caenorhabditis Genetics Center (CGC) provided the RB1550 *rpi-2(K08D12.2)(ok1863)* mutant, and the *rpi-2(ok1863)* allele contains 1,143-bp deletion that removes a large segment of exon III, exon IV, and some portions of exon V. *T05A8.5(syb1568)II.* has 1,888 bp deletions, deleting all exons except exon I (Fig S3B). Sunybiotech created an independent null allele of *elmd-1* via CRISPR–Cas9. The PHX630, *elmd-1(syb630) III.* mutant contains 1,784 bp deletions, where except for exon I, all exons were removed (Fig S2B). *elmd-1(syb630) III.* were outcrossed to WT two times.

### Fluorescent transgenes for IFT proteins

*GOU2162 che-3(cas443[gfp::che-3]) I; xbx-1(cas502[xbx-1::tagRFP]) V; GOU2362 ift-74(cas499[ift-74::gfp])II.; EJP76 vuaSi15 [pBP36; Posm-6:: osm-6::eGFP; cb-unc119(+)] I; unc-119(ed3) III; osm-6(p811) V; N2;lqIs2 [osm-6::gfp], N2;ejEx[osm-3::GFP + pRF4]; N2;ejEx[kap-1::gfp+pRF4]; EJP81.*

vuaSi24 *[pBP43; Pift-140::ift-140::mCherry; cb-unc-119(+)]II; unc-119(ed3) III; ift-140(tm3433) V; jhuEx [ift-140::GFP+pRF4]; Ex[rpi-2:: GFP+ xbx-1::tdTomato+pRF4]; MX76 dpy-5(e907); nxEx(bbs-7::gfp+-dpy-5 (+)).*

### Fluorescent transgenes for ciliary proteins

PHX1180, *wdr-31(syb1180 [wdr-31(T05A8.5)::GFP]); PHX4934 rpi-2(syb4934) [rpi-2::mCherry]; OIK1042 N2;turEx21[arl-13p::wdr-31 (T05A8.5)::GFP::unc-54 3'UTR +rol-6} (5 ng); OIK1044 N2;turEx23 [elmd-1p::GFP::elmd-1 (C56G7.3)::unc-54 3'UTR +rol-6} (5 ng); OIK1045 N2;turEx24[arl-13p::GFP::elmd-1 (C56G7.3)::unc-54 3'UTR + rol-6} (5 ng); OIK1046 N2;turEx25[elmd-1p(C56G7.3)::GFP::unc-54 3'UTR +rol-6} (50 ng); OIK1047 N2;turEx26[wdr-31 (T05A8.5)p::GFP:: unc-54 3'UTR +rol-6} (50 ng); N2;Ex[mks-2::GFP + tram-1::tdTOMATO + pRF4]; MX1409 N2; nxEx785[tax-4:gfp+ Posm-5::xbx-1::tdTomato + rol-6(su1006)]; vuaSi21[pBP39; Pmks-6::mks-6::mCherry; cb-unc-119(+)] II; MX2418 N2;nxEx1259[pbbs-8::PLC-delta PH::GFP;MKSR-2::tdTomato; coel::GFP]; PY8847 oyIs65[str-1p::mcherry]; Ex[str-1p::nphp-4:: gfp, unc122p::dsRed].* All PHX strains were generated using CRISPR/Cas9 by Sunybiotech. The list of extensive transgenic and mutant strains is provided in Table S4.

### Fluorescence and confocal microscopy for *C. elegans*

Fluorescence images (dye assay) and time-lapse movies (IFT movies) were captured using an epi-fluorescence upright microscope (Leica DM6 B) (Video 1, Video 2, Video 3, Video 4, and Video 5) (three frames per second, and up to 120 frames). The

epi-fluorescence upright microscope was controlled with iQ3.6.2 Andor software and was attached with an Andor iXon Ultra 897 EMCCD (an electron-multiplying charge-coupled device camera).

The LSM900 confocal microscope with Airyscan 2 (ZEN three Blue edition software) was used to capture the high-resolution Z-stack images. On the microscope slides, a drop of 2–3% agarose was used to create an agarose pad, and worms were then moved to the agarose pad containing 1–3 $\mu$l of 10 mM levamisole (an anesthetic agent). Images were collected at 0.14 m intervals with a Plan ApoChromat 63x/1.40 NA target, then analyzed with Blue edition software ZEN 3 to create Z-stacks, and processed with ImageJ (NIH) software (Schneider et al, 2012).

### In vivo IFT assay for IFT frequency and velocity

The time-lapse movies of GFP-labelled IFT proteins were analyzed with the automatic kymograph analyzing tools KymographClear and KymographDirect, both of which are ImageJ based (Mangeol et al, 2016; Turan et al, 2022). KymographClear was used to produce kymographs from time-lapse movies. We examined each produced kymograph and calculated IFT frequency or IFT velocities for IFT-74::GFP (an endogenously tagged), GFP::CHE-3 (an endogenously tagged), BBS-7::GFP (an overexpression transgene), OSM-6::GFP (a single copy transgene), OSM-3::GFP (an overexpression transgene), and IFT-140::GFP (an overexpression transgen) in WT, *wdr-31;elmd-1* double and *wdr-31; elmd-1;rpi-2* triple mutants. All worms were maintained at 20°C for IFT analysis. At least 10 IFT videos were taken over three separate time periods for strains, and IFT particles from at least 10 kymographs were counted.

### Whole-mount ISH in zebrafish

Whole-mount ISH was performed as previously described (Thisse et al, 2004). The *wdr31* cDNA with T7 promoter was PCR amplified, with forward primer: ATGGGGAAGCTACAGAGCAAGTTC and reverse primer: TAATACGACTCACTATAGAAGCGAGCCACTTCAGTGATACTG, from a homemade cDNA library of zebrafish embryos at 24 hpf. The antisense probe for *wdr31* was then transcribed with digoxigenin-labeled UTPs and T7 RNA polymerases (Roche). The stained embryos were dehydrated in glycerol and photographed with a Nikon SMZ1500 stereomicroscope (Nikon).

### Mutagenesis of *wdr31* in zebrafish using the CRISPR/Cas9 technology

Generation of zebrafish mutants using the CRISPR/Cas9 system was carried out as previously described (Chang et al, 2013). Briefly, two gRNA-targeting sequences in *wdr31* were chosen as follows: 5′-ACCCATGTGTGTTGGGTACC-3′ and 5′-GAAGCCATCCAGGAGTTCAG-3′. gRNA templates were PCR amplified and gRNAs were in vitro transcribed with T7 transcriptase (cat# M0251S; NEB). gRNAs and Cas9 protein (cat# M0251S; NEB) were simultaneously injected into the embryos at the one-cell stage.

### Immunofluorescence staining and microscopy

Immunostaining was performed as previously described (Xu et al, 2017). Briefly, embryos were fixed in cold Dent's fixative (80% methanol: 20% dimethyl sulfoxide) at –20°C overnight and then stored in methanol. The samples were permeabilized with 0.005% (m/v) trypsin for 30 min. Samples were blocked with blocking buffer (10% [vol/vol] goat serum in PBST), followed by incubation with primary antibody anti–acetylated tubulin (1:2,000; Sigma-Aldrich), and then secondary antibodies conjugated with Alexa Fluor 568 (1: 500; Invitrogen). Samples were mounted on ProLong Gold Antifade Mountant with DAPI (Invitrogen) and images were taken with a Nikon A1R confocal microscope.

### Software and algorithms

Zen Blue Zeiss https://www.zeiss.com/corporate/int/home.html
 Andor iQ3 Andor https://andor.oxinst.com/
 Fiji ImageJ https://fiji.sc
 ImageJ ImageJ https://imagej.nih.gov/ij/
 Illustrator (CS5.1) Adobe https://www.adobe.com/
 KymographClear Peterman Lab https://sites.google.com/site/kymographanalysis
 R R Core Team https://www.r-project.org/.

### Statistical analysis and generating figures

For statistical analysis of dye assay, Fisher's exact test (one tailed tests) was performed. For data involving continuous variables like IFT speed and cilia length, Mann–Whitney *U*-test or Welch's *t* test was used depending on the distribution of data. Transgenic worms were kept at 15°C for AWB cilia morphology analysis and AWB cilia length measurements, and only L4 stage animals were imaged for AWB cilia. For each strain, at least three independent microcopy analyses were performed, and 60–220 AWB cilia were examined. All statistical tests were performed using R software. The codes and files needed to generate figures and perform statistical analysis were openly shared, and the files and codes for making figures and performing statistical analysis may be accessed at https://github.com/thekaplanlab/WDR31-ELMOD-RP2.

### Gateway cloning (Mammalian constructs)

Constructs for ELMOD3 (HsCD00288286) and WDR31 (HsCD00045652) were purchased from Harvard Medical School. LR Reaction (Invitrogen) was performed to transfer the target sequence into destination vectors ((N)RFP, (N)CFP plasmids) with a subsequent transformation into *E. coli* DH5*α*. DNA was isolated according to Monarch Plasmid Miniprep Kit (BioLabs) and the PureYield Plasmid Midiprep Protocol System (Promega). Verification of successful cloning was done by sequencing (Eurofins).

### Immunofluorescence staining (mammals)

hTERT-RPE1 cells were transiently transfected according to Lipofectamine 3000 Reagent Protocol with ELMOD3-CFP and

WDR31-RFP constructs followed by serum starvation for 3 d to induce cilia formation. Cells were fixed with 4% PFA for 45 min at 4°C, permeabilized for 5 min with 0.3% PBST, and blocked with 10% goat serum in PBST at 4°C overnight. A primary antibody for ARL13B (1:50; Proteintech) and a secondary antibody conjugated to Alexa Fluor 647 (1:350; Invitrogen) were used. Cells were mounted using Fluoromount-G (Invitrogen).

### Microscopy setup (mammals)

Images were captured using a Leica TCS SP8 scanning microscope (Leica Microsystems IR GmbH). The setup includes 488 nm, 532 nm, and 635 nm pulsed excitation lasers and 100x oil immersion objective lens (NA 1.4) and a hybrid detector (HyD). Pixel number was 1,024 × 1,024 and optimal pixel size was determined by Nyquist calculation resulting in a size of 43 × 43 nm in XY. In addition, Z-steps should not exceed 131 nm when acquiring a stack. Laser intensity was adjusted for each sample and images were recorded with 2x frame averaging.

### Image processing (mammals)

Acquired images were processed (LasX, Leica Microsystems), deconvoluted (Huygens Software, SVI), and edited (FIJI software). The deconvolution was performed with a Classical Maximum Likelihood Estimation (CMLE) algorithm under experimentally defined settings. The background level was software estimated, the quality threshold was 0.001, the number of iterations was 50 and signal-to-noise ratio was set to 20. All images were brightness-corrected for the purpose of presentation.

### Proteomics analysis

Protein partners for WDR31 and ELMOD3 were investigated. For label-free quantification MaxQuant Software (version 1.6.0.16) was used to identify proteins from MS/MS spectra. The used settings can be found in the parameters file. The results (LFQ intensities) from MaxQuant were uploaded in Perseus software (version 1.6.2.3). Six biological replicates were analysed for the sample (WDR31 and ELMOD3) and the negative control in each experiment was applied a Pearson correlation, which is a measure of the association between two replicates. When one or two replicates had a slightly less correlation compared with the others, we removed this replicate but continued the analysis either way. For analysis, at least four replicates were used. Proteins were removed, that were only identified by site, reverse proteins, and potential contaminants. Among the rest of the proteins, those were disregarded which could not be found in more than half of the samples in at least one group. With these, we performed a two-sample $t$ test (permutation-based FDR, $P$ = 0,05) to see which proteins are significantly enriched compared with the controls, and an outlier test (left sided significance A, Benjamini–Hochberg $P$ = 0,05). The $\log_2$ of the intensity was plotted against the $\log_2$ ratio of the relative protein concentration in the sample to the negative control (scatter plot) (Fig S9).

## Data Availability

All strains and reagents generated for the current study are available from the corresponding author.

## Supplementary Information

## Acknowledgements

We thank Oliver Blacque, Piali Sengupta, Micheal Leroux, the National BioResource Project (NBRP) in Japan, and the CGC in the United States, which is financed by the NIH Office of Research Infrastructure Programs (P40 OD010440) for sharing valuable strains. We thank Atiyye Zorluer for generating strains. We thank Oliver Blacque and Samuel Katz for critical reading of the manuscript. We thank the Abdullah Gul University Scientific Research Project Coordination Unit (Project number: TOA-2018-110) for providing the funding that initiated the project. M Ueffing was supported by the Tistou & Charlotte Kerstan Stiftung. The Leica laser scanning microscope was funded by a grant from Deutsche Forschungsgemeinschaft (INST 2388/62-1).

### Author Contributions

S Cevik: resources, data curation, formal analysis, validation, investigation, visualization, methodology, and writing—review and editing.
X Peng: formal analysis, validation, and methodology.
T Beyer: formal analysis, validation, and methodology.
MS Pir: formal analysis and methodology.
F Yenisert: formal analysis and methodology.
F Woerz: formal analysis, validation, and methodology.
F Hoffmann: formal analysis, validation, and methodology.
B Altunkaynak: formal analysis and methodology.
B Pir: formal analysis and methodology.
K Boldt: formal analysis, validation, and methodology.
A Karaman: formal analysis and methodology.
M Cakiroglu: formal analysis and methodology.
SS Oner: formal analysis, validation, and methodology.
Y Cao: formal analysis, validation, and methodology.
M Ueffing: formal analysis, validation, and methodology.
OI Kaplan: conceptualization, resources, data curation, software, formal analysis, supervision, funding acquisition, validation, investigation, visualization, methodology, project administration, and writing—original draft, review, and editing.

### Conflict of Interest Statement

The authors declare that they have no conflict of interest.

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
