## [Reviewer comments · Life Science Alliance]

Life Science Alliance

Functional redundancy of WDR31 with ELMOD and RP2 in regulating IFT trains via BBSome

Sebiha Cevik, Xiaoyu Peng, Tina Beyer, Mustafa Pir, Ferhan Yenisert, Franziska Woerz, Felix Hoffmann, Betul Altunkaynak, Betul Pir, Karsten Boldt, Asli Karaman, Miray Cakiroglu, S. Sadik Oner, Ying Cao, Marius Ueffing, Oktay Kaplan
DOI: <https://doi.org/10.26508/lsa.202201844>

Corresponding author(s): Dr. Oktay I. Kaplan (Abdullah Gül University)

Review Timeline:

Submission Date:	2022-11-21
Editorial Decision:	2023-01-09
Revision Received:	2023-04-06
Editorial Decision:	2023-05-03
Revision Received:	2023-05-09
Accepted:	2023-05-09

Scientific Editor: Novella Guidi

Transaction Report:

January 9, 2023

Re: Life Science Alliance manuscript #LSA-2022-01844-T

Oktay I Kaplan
School of Biomolecular and Biomedical Science
Dublin Dublin 4
Ireland

Dear Dr. Kaplan,

Thank you for submitting your manuscript entitled "Functional redundancy of WDR31 with GTPase-activating proteins ELMOD and RP2 in regulating IFT trains via BBSome" to Life Science Alliance. The manuscript was assessed by expert reviewers, whose comments are appended to this letter. We invite you to submit a revised manuscript addressing the Reviewer comments.

Thank you for this interesting contribution to Life Science Alliance. We are looking forward to receiving your revised manuscript.

Sincerely,

B. MANUSCRIPT ORGANIZATION AND FORMATTING:

Reviewer #1 (Comments to the Authors (Required)):

This manuscript describes the involvement of WDR31 in cilia formation in zebra fish and *C. elegans*. The authors go on to show that in the nematode, WDR31 may genetically interact with ELMD1 and RP2, two GTPase activating proteins to modulate cilia formation, IFT speed and maintaining barrier integrity of the cilia. There are some concerns regarding the study:

1. How do the authors explain that the phenotypes of the triple mutants can be rescued by expressing any of the three genes.
2. The authors should examine a potential physical interaction between the three proteins to validate the genetic interaction model.
3. If the phenotypes between zebra fish and nematodes were strikingly different, it is important to validate these findings in a mammalian model/platform.

Reviewer #2 (Comments to the Authors (Required)):

The study from the Kaplan lab identified WDR31 as a protein localizing at the ciliary base. They used zebrafish and *Caenorhabditis elegans* to show the function WDR31 in cilia length and morphology cilia morphology. They found that, in a triple knockout strain deleting WDR-31, RP-2, ELMD-1, fewer IFT/BBSome particles travel along cilia. They further showed that the anterograde IFT in the middle segment accelerates and a non-ciliary protein abnormally enters cilia in *wdr-31;rpi-2;elmd-1*. They suggest that WDR31-RP-2-ELMD-1 regulates IFT and BBSome trafficking. Overall, this is an interesting study and worth publication in this journal as long as the follow issues can be resolved. Ideally, some additional experiments can certainly help with the revision. However, the Reviewer understands the difficult current situation for each lab. If the authors cannot perform experiments, they should at least carefully analyze their existing data and acknowledge the limitation.

First of all, what happened to the ciliary length in the triple mutants. In figure 4F, they showed that ciliary length is reduced using a diffusive GFP reporter; however, in figure 5A, they used IFT-markers for PHA/PHB, from the eye of the reviewer, the ciliary length is actually in the single, double or triple mutant than wild type. TEM will provide a solid description. If the author has such data, they should include them. Otherwise, the authors need to discuss the differences.

The authors used kymography to quantify IFT, which is good. However, they should follow the published standard to show the histogram of the speeds. Through such careful quantification, the authors may reveal additional mechanistic insights. Considering that the authors isolate three proteins localizing at the ciliary base affects IFT, mechanistic insights remain limited.

Reviewer Comments to Authors: (Response to Reviewer Comments were labeled red)**Reviewer: 1**

1.1 How do the authors explain that the phenotypes of the triple mutants can be rescued by expressing any of the three genes.

Response: With the reintroduction of either ELMD-1 or RPI-2, we also observed the restoration of triple mutant phenotypes; however, we did not include a comment in order to keep the paper's emphasis intact. It appears that ELMD-1 or RPI-2 overexpression can substitute one another. A part of reason might be is that they both might regulate the same GTPases even though this is currently unknown and might be a great research question for the future. As RP2 is a GTPase activating protein (GAP) for ARL3 and ARL3 activation may be the cause of these abnormalities, we initially considered introducing arl-3 mutants, but creating quadruple mutants would take a lot of efforts.

1.2. The authors should examine a potential physical interaction between the three proteins to validate the genetic interaction model.

Response: Due to unexpected COVID symptoms in January, our collaborator was unable to execute their responsibilities, but could provide proteomics analysis of WDR31 and ELMOD3, which are added as a supplementary figures and files. Please see Suppl. Figure 9 and Table S4. In the studied experimental settings, neither of these two proteins was brought down by the other, but we cannot rule out a transient and temporary interaction between them. We added all relevant information in the Line 639-640 in the discussion and provided explanation for materials and methods under the name of Proteomics Analysis (Line 320-336).

1.2. If the phenotypes between zebraish and nematodes were strikingly different, it is important to validate these findings in a mammalian model/platform.

Response: Both the Wdr31 deficient zebrafish and the wdr-31 null mutant nematode have cilia that are normal in length, although some cilia are missing in the mutant zebrafish lacking Wdr31. Although the obvious cause is not yet identified, the reviewer suggested that knocking out WDR31 in animals will enable us to comprehend differences. We fully agree with the reviewer. As noted above, Due to unexpected COVID symptoms in January, our collaborator was unable to execute their responsibilities. We are sorry about it. We added the relevant explanation into discussion, starting line 607.

Reviewer: 2

1.2. First of all, what happened to the ciliary length in the triple mutants. In figure 4F, they showed that ciliary length is reduced using a diffusive GFP reporter; however, in figure 5A, they used IFT-markers for PHA/PHB, from the eye of the reviewer, the ciliary length is actually in the single, double or triple mutant than wild type. TEM will provide a solid description. If the author has such data, they should include them. Otherwise, the authors need to discuss the differences.

Response: We really appreciate the suggestion from the reviewer. Because certain IFT markers (IFT-140/CHE-11) do not penetrate the distal segment of cilia and because there are distal segment accumulations of IFT protein (OSM-3, IFT-74) in the triple mutants, we did not employ the IFT-based markers for cilia length measurements. Therefore, we choose a diffusive GFP reporter. The average PHA/PHB cilia length was apparently shorter in the triple mutants, meaning that there are longer and shorter cilia in the triple mutants, according to a diffusive GFP reporter. As suggested by the reviewer, TEM would be best to understand to resolve the issue. Despite our desire to respond experimentally, a number of factors (finance and earthquakes) precluded us from doing so. Due to unexpected, recent earthquakes (7.8 and 7.5) that struck the region nearby have left us struggling with the effects. During the second major earthquake (a 7.5), we were in the lab, and one of the ceiling panels there collapsed as

a result of the earthquake and fell to the ground. Fortunately, we were able to leave the lab before one of the ceiling panels collapsed. We had stopped working in the lab and had additional panels removed in order to protect each of us from further harm.

1.2. The authors used kymography to quantify IFT, which is good. However, they should follow the published standard to show the histogram of the speeds. Through such careful quantification, the authors may reveal additional mechanistic insights. Considering that the authors isolate three proteins localizing at the ciliary base affects IFT, mechanistic insights remain limited.

Response: We thank the reviewer for the suggestion. As suggested, we re-formatted the data and used both line plot and histogram to show the distribution of data. (Figure 8 and Figure S9). The relevant changes in the text was made. Line 507-509 and Line 563-566, Line 811-815, and Line 905-907

May 3, 2023

RE: Life Science Alliance Manuscript #LSA-2022-01844-TR

Dr. Oktay I Kaplan
Abdullah Gül University
School of Life and Natural Sciences,
Abdullah Gul University,
Kayseri 38090
Turkey

Dear Dr. Kaplan,

Thank you for submitting your revised manuscript entitled "Functional redundancy of WDR31 with ELMOD and RP2 in regulating IFT trains via BBSome". We would be happy to publish your paper in Life Science Alliance pending final revisions necessary to meet our formatting guidelines.

-please consult our manuscript preparation guidelines <https://www.life-science-alliance.org/manuscript-prep> and make sure your manuscript sections are in the correct order
-please use the [10 author names, et al.] format in your references (i.e. limit the author names to the first 10)

Figure Check:

-please add scale bars to Fig. 3A

A. FINAL FILES:

B. MANUSCRIPT ORGANIZATION AND FORMATTING:

Sincerely,

Reviewer #1 (Comments to the Authors (Required)):

The authors have satisfactorily addressed the concerns.

May 9, 2023

RE: Life Science Alliance Manuscript #LSA-2022-01844-TRR

Dr. Oktay I Kaplan
Abdullah Gül University
School of Life and Natural Sciences,
Abdullah Gul University,
Kayseri 38090
Turkey

Dear Dr. Kaplan,

Thank you for submitting your Research Article entitled "Functional redundancy of WDR31 with ELMOD and RP2 in regulating IFT trains via BBSome". It is a pleasure to let you know that your manuscript is now accepted for publication in Life Science Alliance. Congratulations on this interesting work.

DISTRIBUTION OF MATERIALS:

Again, congratulations on a very nice paper. I hope you found the review process to be constructive and are pleased with how the manuscript was handled editorially. We look forward to future exciting submissions from your lab.

Sincerely,
